# Developmental disruption and restoration of brain synaptome architecture in the murine Pax6 neurodevelopmental disease model

Laura Tomas-Roca[1], Zhen Qiu[1], Erik Fransén [2], Ragini Gokhale[1], Edita Bulovaite[1], David J. Price[3], Noboru H. Komiyama[1,3] & Seth G. N. Grant [1,3] ✉

Neurodevelopmental disorders of genetic origin delay the acquisition of normal abilities and cause disabling phenotypes. Nevertheless, spontaneous attenuation and even complete amelioration of symptoms in early childhood and adolescence can occur in many disorders, suggesting that brain circuits possess an intrinsic capacity to overcome the deficits arising from some germline mutations. We examined the molecular composition of almost a trillion excitatory synapses on a brain-wide scale between birth and adulthood in mice carrying a mutation in the homeobox transcription factor *Pax6*, a neurodevelopmental disorder model. *Pax6* haploinsufficiency had no impact on total synapse number at any age. By contrast, the molecular composition of excitatory synapses, the postnatal expansion of synapse diversity and the acquisition of normal synaptome architecture were delayed in all brain regions, interfering with networks and electrophysiological simulations of cognitive functions. Specific excitatory synapse types and subtypes were affected in two key developmental age-windows. These phenotypes were reversed within 2-3 weeks of onset, restoring synapse diversity and synaptome architecture to the normal developmental trajectory. Synapse subtypes with rapid protein turnover mediated the synaptome remodeling. This brain-wide capacity for remodeling of synapse molecular composition to recover and maintain the developmental trajectory of synaptome architecture may help confer resilience to neurodevelopmental genetic disorders.

Excitatory synapses, which make up the vast majority of synapses in the brain, have highly diverse identities resulting from their differing protein composition and protein lifetimes[1–3]. The anatomical distribution of these molecularly diverse synapses can be studied using synaptome mapping technology, a large-scale, single-synapse resolution, systematic image analysis approach that has uncovered brain-wide excitatory synapse diversity in the mouse[1–3]. The Mouse Lifespan Synaptome Atlas[2] revealed that the synapse composition of the brain changes continuously across the lifespan, with trajectories of

excitatory synapse types and subtypes in dendrites, neurons, circuits and brain regions, together defining the Lifespan Synaptome Architecture (LSA)[2]. A key feature of the LSA is its three distinct age-windows (LSA-I, LSA-II, LSA-III), which correspond to childhood, adolescence/young adulthood, and the aging adult, respectively. During LSA-I, which extends from birth until weaning, there is a dramatic increase in synapse number and synapse diversity. Following the transition to independence from maternal care, the synaptome architecture of LSA-II progressively develops until adult sexual maturity with further

[1]Genes to Cognition Program, Centre for Clinical Brain Sciences, University of Edinburgh, Edinburgh EH16 4SB, UK. [2]Science for Life Laboratory, KTH Royal Institute of Technology, SE-171 65 Solna, Sweden. [3]Simons Initiative for the Developing Brain (SIDB), Centre for Discovery Brain Sciences, University of Edinburgh, Edinburgh EH8 9XD, UK. ✉e-mail: seth.grant@ed.ac.uk

increases in synapse diversity and differentiation of the synaptome composition of brain regions[2].

Although the genetic mechanisms controlling synaptome architecture are only beginning to be understood, analysis of the spatial patterning of synapses suggests that developmental mechanisms controlling the body plan might be involved[3]. Transcription factors containing homeodomains play a key role in establishing the body plan and in development of the brain[4–10] and we hypothesized that they might control synaptome architecture. Pax6, a member of the homeodomain transcription factor family, regulates the expression of many synaptic proteins including PSD95 and SAP102[11], two postsynaptic proteins in excitatory synapses that are used in synaptome mapping[2,3]. Heterozygous (haploinsufficient) loss-of-function *PAX6* mutations cause autism, intellectual disability, epilepsy and aniridia (WAGR syndrome)[12–19], which are phenocopied in *Pax6*[+/−] mice[18,20–23]. Pax6 is expressed during embryogenesis in progenitor cells giving rise to forebrain and hindbrain structures, and postnatally in subsets of diencephalic neurons[24–27]. In this study, we have characterized the effect of a germline mutation in *Pax6* on early and late postnatal development (LSA-I to mid LSA-II) of the mouse brain synaptome architecture. Our findings not only show that the development of brain synaptome architecture is organized by this homeobox gene, but also that the synaptome has the capacity to repair itself during phases of mouse development representing childhood and adolescence.

## Results

### Mapping the developing synaptome architecture of *Pax6*[+/−] mice

The synaptome architecture of *Pax6*[+/−] mice was analyzed using the SYNMAP synaptome mapping pipeline[2,3], which quantifies the expression of postsynaptic proteins PSD95 and SAP102 in synaptic puncta (Fig. 1). PSD95 and SAP102 are two of the four paralogous members of the MAGUK family and their levels are functionally important because they physically assemble multiple proteins controlling synaptic transmission, plasticity and neuronal excitability into macromolecular complexes[28–31], and altering their expression leads to changes in synaptic and cognitive functions[32,33]. Previously, we have characterized mice carrying engineered alleles of endogenous PSD95 and SAP102 in a wide range of biochemical, anatomical, physiological and behavioral studies[1–3,28–30,32–41]. Here we used mice that have been engineered to express the fluorescent proteins eGFP and mKO2 attached to the carboxyl terminus of PSD95 and SAP102, respectively, which enable visualization of synaptic puncta[2,3]. *Pax6*[+/−] mice were crossed with PSD95-eGFP and SAP102-mKO2 mice to generate cohorts of *Pax6*[+/−];*Psd95*[eGFP/eGFP];*Sap102*[mKO2/mKO2] and control *Pax6*[+/+];*Psd95*[eGFP/eGFP];*Sap102*[mKO2/mKO2] mice. Parasagittal brain sections were collected at day one (P1) and at one (P7), two (P14), three (P21), four (P28), five (P35), six (P42), seven (P49) and eight (P56) weeks. The first five time points correspond to LSA-I (P1-P28) and the latter four to LSA-II (P35-P56)[2].

Brain sections were imaged at single-synapse resolution on a spinning disk confocal microscope (pixel size 84 × 84 nm and optical resolution ~260 nm) and the density, intensity, size and shape parameters of individual puncta were acquired in 131 brain subregions. We classified the synaptic puncta into three types (type 1 express PSD95 only, type 2 express SAP102 only, and type 3 express both PSD95 and SAP102), which were classified into 37 subtypes on the basis of morphological (size and shape) features[3] (type 1 subtypes 1–11; type 2 subtypes 12–18; type 3 subtypes 19–37). All data were registered to the Allen Developing Mouse Brain Atlas (Supplementary Data 1) and are available in Supplementary Data 2–16 and at Edinburgh DataShare[42]. We created the Pax6 Developmental Synaptome Atlas[43], an interactive visualization tool for displaying the spatial framework of datasets and differences in the developing brain synaptome of control and *Pax6*[+/−] mice.

### *Pax6*[+/−] mice show transient synaptome phenotypes in two developmental age-windows

We measured a total of $3.65 \times 10^{11}$ excitatory synapses and found no significant differences in synapse number between *Pax6*[+/−] and control mice in any brain region or in the whole brain at any age ($P > 0.05$, Benjamini−Hochberg corrected). However, quantifying the synapse types and subtypes at each of the nine ages (Figs. 2 and S1) indicated that the molecular composition of excitatory synapses was substantially impacted by this germline homeobox gene mutation. At birth (P1) the *Pax6*[+/−] synaptome was largely normal, but during the second and third postnatal weeks (P7–P21) strong synaptome phenotypes emerged in most brain regions, which then reverted to normal in week four (P28). Synaptome phenotypes then remerged in week five (P35–P42) before reverting again to normal by P49. This temporal progression of synaptome phenotypes describes two 'phenotype waves', one starting in the second postnatal week and the other in the sixth postnatal week, each lasting approximately two weeks. At the peak of each phenotype wave, synapse composition was affected in almost every region and subregion of the *Pax6*[+/−] brain (Figs. 2 and S1). The strongest phenotypes were observed in the neocortex, including visual cortex (Figs. 2, S1, Supplementary Data 16).

We next examined the impact of the *Pax6*[+/−] mutation on excitatory synapse type and subtype diversity. As shown in the summary plots (Figs. 2 and S1) and example single-synapse resolution images (Fig. 3), synapse types and subtypes were differentially affected during the phenotype waves. In both waves there was a loss of type 2 synapses (which express SAP102 only) and an increase in type 1 and 3 synapses (which express PSD95), suggesting that the PSD95-expressing synapses are compensating or adapting to the loss of SAP102-expressing synapses. Looking at the synapse composition of each brain subregion at the different ages (Fig. 4A) revealed a lower synapse diversity across most regions of the *Pax6*[+/−] brain compared with control brains during the first phenotype wave (diversity was reduced in 34/131 subregions in P1, 117/131 in P7 and 63/131 in P14). These results indicate that the molecular identity of synapse types and subtypes is affected during brain development in *Pax6*[+/−] mice, resulting in a lower synapse diversity during the phenotype wave in LSA-I. As the animals aged, the synapse composition of brain regions recovered to the normal developmental trajectory and synapse diversity increased to control levels.

We recently found that PSD95-expressing excitatory synapse subtypes differ greatly in their rate of protein turnover, indicating that some subtypes can more rapidly remodel their proteomes[1]. We hypothesized that these synapse subtypes could be capable of mediating the phenotype waves observed in *Pax6*[+/−] mice. To address this hypothesis, we ranked the 30 PSD95-expressing synapse subtypes and identified those that were widely impacted throughout the *Pax6*[+/−] brain (Fig. 4B). Short protein lifetime (SPL) (subtypes 6, 8, 11, 28, 29, 31) and long protein lifetime (LPL) (subtypes 2, 3, 5, 30, 34) synapses had skewed distributions, with SPL synapses disrupted in more brain regions than LPL synapses. SPL synapses were disproportionately affected in *Pax6*[+/−] mice, showing a significantly stronger phenotype than LPL synapses ($P < 0.05$, Bayesian test with Benjamini−Hochberg correction) in 72% (94/131) of subregions at P14 and in 28% (36/131) of subregions at P35 (Fig. 4B, C). By contrast, only 3% (4/131) of subregions at P14 and 7% (9/131) of subregions at P35 showed a significantly stronger phenotype in LPL over SPL synapses ($P < 0.05$, Bayesian test with Benjamini-Hochberg correction) (Fig. 4B, C). These results show that the dynamic and transient nature of synaptome phenotypes resides largely with SPL synapses, whereas LPL synapses remain significantly more resilient to the impact of the *Pax6*[+/−] mutation.

### *Pax6* mutation transiently disrupts brain network structure and function

The synaptome architecture of the brain can be described as a network of brain regions[3], which has previously been shown to correlate with

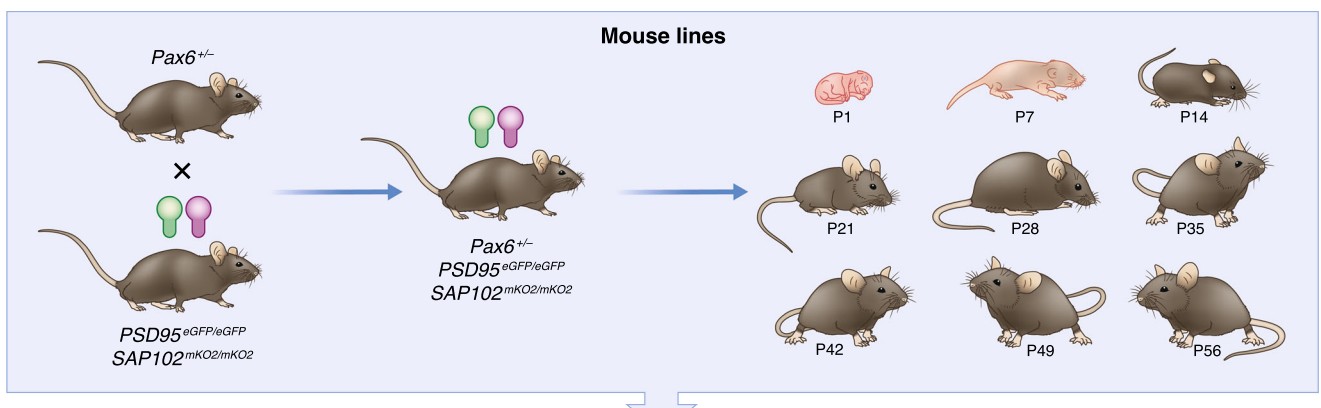

**Fig. 1 | Procedure for mapping the synaptome architecture of *Pax6*⁺/⁻ mice.** Mice carrying the heterozygous *Pax6* mutation were crossed with mice carrying fluorescently tagged PSD95 and SAP102 proteins to produce cohorts of *Pax6*⁺/⁻;*Psd95*^eGFP/eGFP^;*Sap102*^mKO2/mKO2^ and control *Pax6*⁺/⁺;*Psd95*^eGFP/eGFP^;*Sap102*^mKO2/mKO2^ mice. Genetic tagging of the endogenous *Psd95* and *Sap102* loci with eGFP and mKO2 results in the expression of PSD95-eGFP and SAP102-mKO2 fluorescent fusion proteins, which assemble into postsynaptic protein complexes. The differential distribution of these proteins into synapses underlies synapse type diversity, which can be visualized in brain tissue sections by spinning-disk confocal microscopy. In the SYNMAP computational pipeline, raw images of fluorescent synaptic puncta are detected, segmented, and the density, intensity, size and shape of each punctum determined. The puncta are then classified into three types and 37 subtypes based on their molecular and morphological parameters. To create the Pax6 Developmental Synaptome Atlas[43], the parameters of these synapse types and subtypes were quantified in 131 brain subregions and their spatial maps and temporal trajectories presented.

structural connectome networks and with dynamic network activity measured using resting-state fMRI[3]. To assess the impact of the *Pax6*⁺/⁻ mutation on brain synaptome networks, we examined the similarity of the synaptome between all brain regions at each age and the topology of networks built from the similarity matrices (Figs. 5 and S2). The similarity matrices of control mice at birth show high and homogeneous similarity between brain regions, followed by a rapid decline in similarity by P14, consistent with previous results[2]. By contrast, in *Pax6*⁺/⁻ mice this decline was delayed, resulting in major differences between the similarity matrices and, therefore, in the similarity ratio

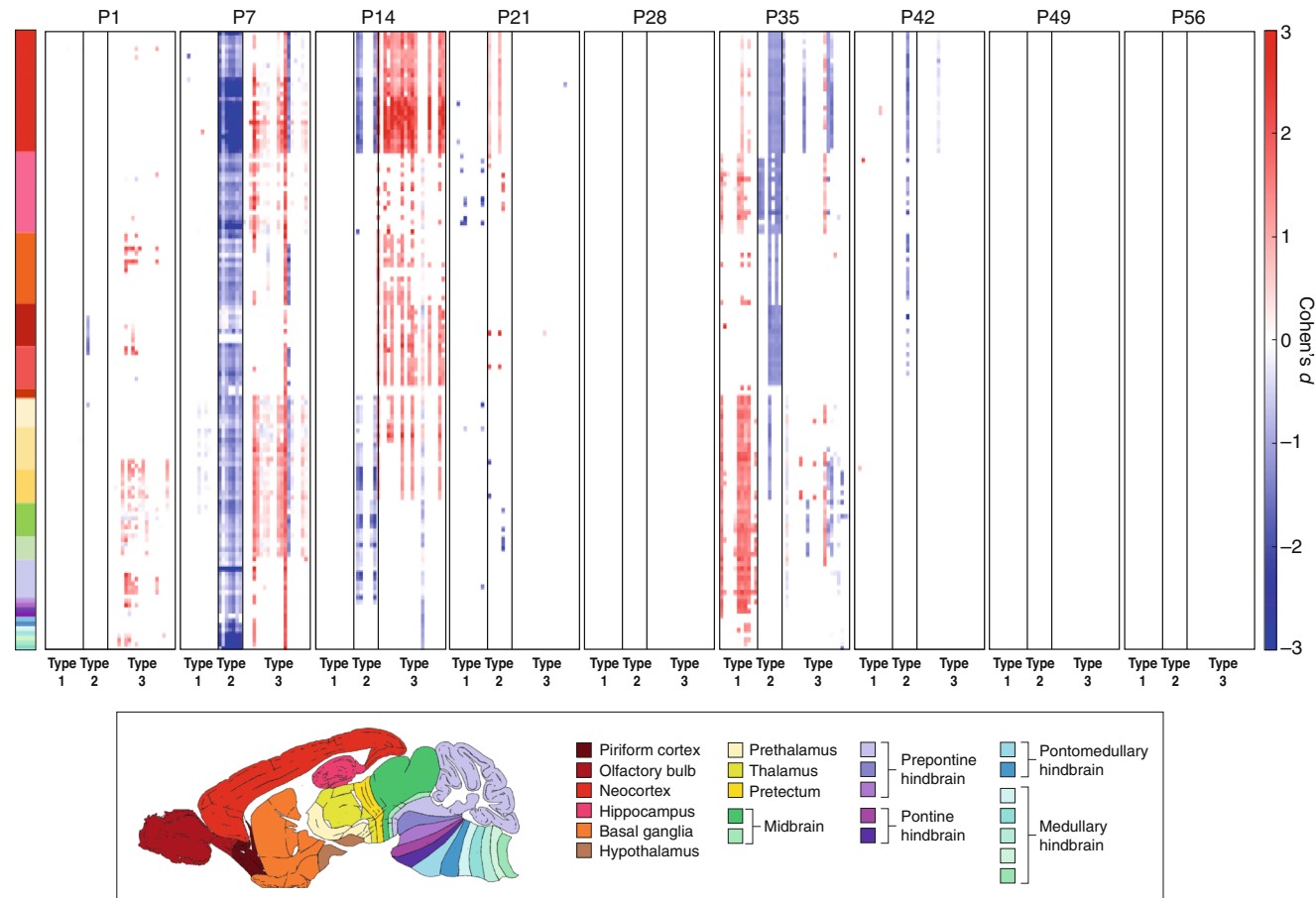

**Fig. 2 | Brain synaptome phenotypes during postnatal development of *Pax6*[+/−] mice.** The difference (Cohen's *d*) in synapse type and subtype density in 131 brain subregions at nine ages from birth to P56 between *Pax6*[+/−] and control mice. Significantly different (*P* < 0.05, Bayesian test, Benjamini–Hochberg corrected) subregions are shown and colored according to effect size (Cohen's *d* scale bar). High-resolution graphs are provided in Fig. S1.

compared with control mice at P14 (Cohen's *d* = 4.27 and *P* < 0.005, Bayesian test with Benjamini–Hochberg correction, Figs. 5A, B and S2). This developmental delay was overcome in the following week (*P* > 0.05, Bayesian test with Benjamini–Hochberg correction, Figs. 5A, B and S2), corresponding to the end of LSA-I. Two weeks later, at P35 (in LSA-II), there was another significant increase in the similarity of brain regions in *Pax6*[+/−] mice (Cohen's *d* = 1.24 and *P* < 0.05 in similarity ratio, Bayesian test with Benjamini-Hochberg correction, Figs. 5A, B and S2), which again was followed by a return to control values in the subsequent two weeks. When we examined the topology of synaptome networks using an index of small worldness[2,3], this showed major reductions in small worldness at P14 (Cohen's *d* = −3.14 and *P* < 0.05, Bayesian test with Benjamini–Hochberg correction, Fig. 5C) and P35 (Cohen's *d* = −4.48 and *P* < 0.05, Bayesian test with Benjamini–Hochberg correction, Fig. 5C) in the *Pax6*[+/−] brain. These results indicate that the network properties of the brain are transiently impaired during the two phenotype waves.

To explore the functional consequences of these synaptome architectural phenotypes in terms of synaptic electrophysiological properties relevant to the storage and recall of behavioral representations, we employed a computational simulation approach[2,3]. In the model, synapses in the CA1 stratum radiatum contain different amounts of PSD95 and SAP102 (Figs. 2, S1, and S3) and their respective electrophysiological responses are simulated using parameters from previous studies of PSD95 and SAP102 in synaptic transmission and plasticity[33,44–47]. The model, which does not require a complete understanding of the molecular mechanisms by which PSD95 and SAP102 regulate AMPA receptors, can address the question of how

different temporal patterns are integrated based on the level of synaptic spatial diversity and, specifically, whether this integration is sensitive to the temporal pattern. In the simulation, the synaptome of CA1 pyramidal neurons is stimulated with patterns of neural activity and the spatial output of excitatory postsynaptic potentials (EPSPs) is quantified at ages before, during and after the two phenotype waves (P1, P7, P28, P35 and P56) (Figs. 6A and S4, Supplementary Data 14, 15). We found that theta burst and gamma train stimulation (Fig. 6B), but not theta train or gamma burst (Fig. S4), resulted in significant phenotypes at P7 and P35 in *Pax6*[+/−] mice (*P* < 0.01, paired *t*-test, Benjamini–Hochberg corrected, *N* = 121 samples). These findings indicate that the synaptome in the CA1 region, which is a structure crucial for spatial navigation, learning and memory[48], is reversibly impaired during the two phenotype waves in *Pax6*[+/−] mice. Moreover, the phenotypes are restricted to particular patterns of neuronal activity.

## Discussion

We have uncovered the effects of a germline mutation in *Pax6*, a homeobox transcription factor, on the postnatal development of synaptome architecture of the mouse brain. We analyzed almost a trillion individual excitatory synapses at weekly intervals from birth to maturity at 2 months of age, creating the Pax6 Developmental Synaptome Atlas[43]. The total number of excitatory synapses was not affected in any brain region at any age in *Pax6*[+/−] mice. Instead, there were substantial changes in the regional composition of excitatory synapse types and subtypes. These synaptome phenotypes emerged transiently in two age-windows,

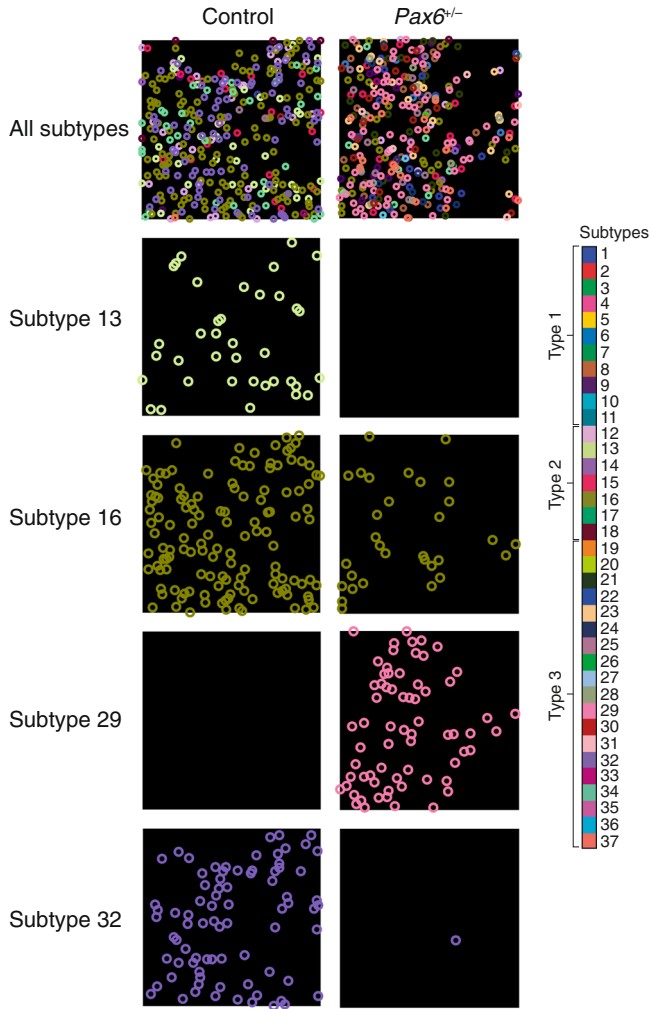

Control          Pax6+/−

All subtypes

Subtype 13

Subtype 16

Subtype 29

Subtype 32

Subtypes
Type 1
Type 2
Type 3

**Fig. 3 | Synapse composition is altered during the synaptome phenotype waves.** Exemplar puncta spatial distributions of synapse subtypes in single tile images (area 21 × 21 μm) of P7 control (left) and *Pax6+/−* (right) mouse brain. Whereas the overall puncta density is unchanged, individual synapse subtypes, and therefore the overall subtype composition, are significantly affected by the *Pax6+/−* mutation.

embryonic and postnatal brain development. During early neurulation, Pax6 is expressed by progenitor cells located throughout much of the neural plate and neural tube, including regions that form the forebrain, hindbrain and spinal cord but not those that form the midbrain[18,25,27,52–54]. Pax6 expression disappears from many of these regions as neurogenesis proceeds, although it is retained by cells in some areas including the olfactory bulb, amygdala, thalamus and cerebellum[18]. Pax6 regulates a wide range of genes including those encoding PSD95, SAP102 and other synaptic proteins such as AMPH, NRXN3, SYNGAP1, SYNPR, SYT11 and SYT17[11], and is expressed postnatally in diencephalic and cerebellar neurons, where we observed strong phenotypes at P7 and P14 in type 2 synapses and at P35 in type 1 synapses. Pax6 is also required for postnatal neurogenesis in the hippocampus[55], which may also contribute to the observed synaptome phenotypes. It is likely, therefore, that the abnormalities in synaptome architecture reported here could arise from both the reduction in Pax6 expression at postnatal ages as well as from residual defects in neurons whose lineages no longer express Pax6. During a critical period of early development, Pax6 protects early progenitors from erroneous specification by inductive signals and thereby controls the identity of neurons long after Pax6 has ceased to be expressed[56]. Neurons that would be expected to have cell-autonomous defects in *Pax6+/−* mice project their axons to neurons that do not express Pax6, such as midbrain neurons, and these postsynaptic neurons might alter their PSD95 and SAP102 expression as a result of the mutation-dependent changes in neuronal activity in the input neurons. Consistent with this, Vitalis and colleagues have shown that loss of Pax6 in populations of neurons causes cell non-autonomous phenotypes in populations of neurons whose lineages have never expressed Pax6[52].

The age-windows during which synaptome architecture was restored in *Pax6+/−* mice correspond to two important transitions in the life of a mouse: from dependency on the mother to independent feeding in LSA-I, and the attainment of sexual maturity and adult behaviors in LSA-II. By ensuring normal trajectories of synaptome architecture during these crucial transitions, maladaptive behaviors caused by underlying genetic variation would be minimized and the mice more likely to survive. The brain of young animals is highly enriched in SPL synapses[1], providing a pool of synapse subtypes capable of rapidly remodeling and repairing the synaptome architecture during development.

In humans, neurodevelopmental disorders delay the acquisition of speech and language, social interactions, learning, attention and motor skills, and also manifest with the onset of epilepsy or motoric dysfunction. Despite this, spontaneous attenuation and even complete amelioration of symptoms occur in early childhood and adolescence in some individuals[57–66]. This amelioration affects behaviors controlled by different brain regions and arising from diverse types of mutation, indicating that the capacity for spontaneous recovery from neurodevelopmental disorders is a brain-wide and general mechanism in the developing nervous system.

Eighty years ago Waddington introduced the concept of canalization as a mechanism that maintains normal trajectories of development in the face of genetic mutations and environmental perturbations[67]. Canalization has been invoked as an explanation for why apparently normal individuals carry deleterious mutations, why symptom penetrance varies in diseases[35,64–73], and how developing neuronal networks overcome mutations in vitro[35]. Our results suggest that remodeling the molecular composition of synapses is a mechanism of canalization in the developing brain. Synaptome canalization does not completely mask all mutations because adult mice lacking *Dlg2* (*Psd93*) or *Dlg3* (*Sap102*) have abnormal synaptome architecture[3]. It is conceivable that therapeutic approaches, potentially targeting SPL synapse subtypes, could enhance remodeling of the synaptome and induce resilience to neurodevelopmental disorders and environmental insults.

indicating that the molecular identity of excitatory synapses is not only dynamic but also has the capacity to restore a normal developmental trajectory of brain synaptome architecture in a genetic neurodevelopmental disorder.

The restoration of synaptome architecture required changes in synapse type and subtype molecular diversity, indicating that their postsynaptic proteomes were being remodeled. Protein turnover is a fundamental mechanism for homeostatic maintenance of the proteome (proteostasis) and is required for proteome remodeling instructed by transcriptional programs[49–51]. Consistent with this, in *Pax6+/−* mice it was the density of synapse subtypes with the fastest protein turnover (SPL synapses) that was most affected during the age-windows. Furthermore, the half-life of PSD95 in SPL synapses is around one to two weeks[1], which corresponds well with the duration of synaptome repair in *Pax6+/−* mice. These results, together with the dynamic changes in the synaptome and in the turnover of synaptic proteins across the lifespan[1,2], indicate that the capacity to remodel the molecular identity of synapses is a property of all regions of the brain.

The synaptome architecture phenotypes are likely to arise from both cell-autonomous and cell non-autonomous mechanisms during

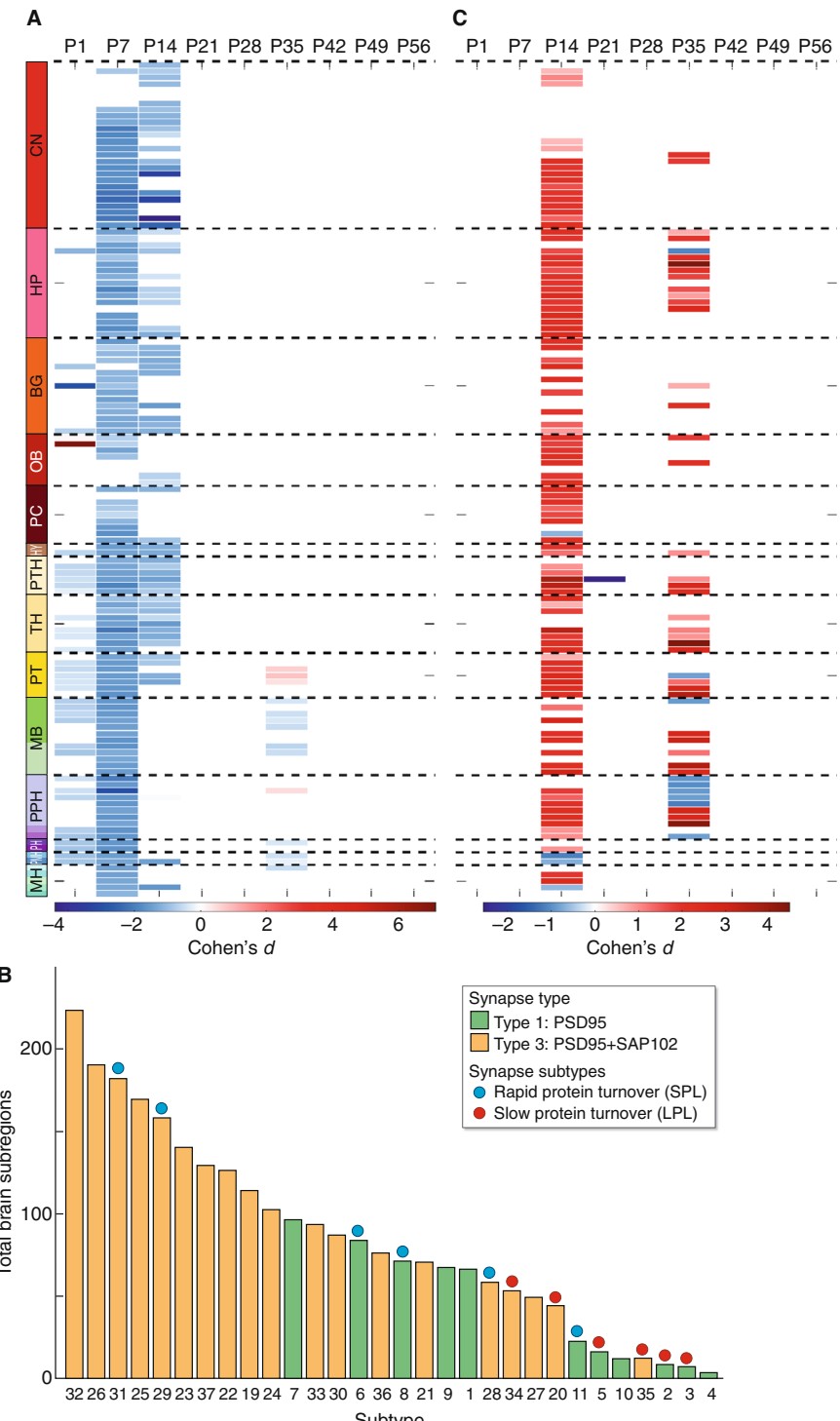

**Fig. 4 | Synapse subtypes with rapid protein turnover are preferentially affected in the *Pax6*⁺ᐟ⁻ brain. A** Changes in synapse diversity in brain subregions between *Pax6*⁺ᐟ⁻ and control mice for the different age groups. Scale bar (Cohen's *d*) indicates increase in red and decrease in blue (*P* < 0.05, Bayesian test with Benjamini−Hochberg correction) of synapse diversity in *Pax6*⁺ᐟ⁻ brain subregions. Absence of color indicates subregions where no significant differences occurred. **B** Ranking of PSD95-expressing synapse subtypes according to the number of brain regions affected in *Pax6*⁺ᐟ⁻ mice. For each subtype the total number of brain subregions that showed a significant difference (*P* < 0.05, Bayesian test with Benjamini−Hochberg correction) in density between control and *Pax6*⁺ᐟ⁻ mice was counted at all ages. Note skewing in SPL and LPL subtypes. Source data are provided as a Source Data file. **C** Quantification of the relative effect of the *Pax6*⁺ᐟ⁻ mutation on SPL and LPL synapses. Red indicates a greater effect on SPL synapses and blue indicates a greater effect on LPL synapses. Scale bar, Cohen's *d*; *P* < 0.05, Bayesian test with Benjamini-Hochberg correction. For brain region abbreviations (**A**, **C**) see Supplementary Data 1.

Synaptome mapping is a highly scalable technology that has enabled systematic analysis of the number and molecular properties of excitatory synapses on a brain-wide scale throughout development in a neurodevelopmental disorder. To understand the cellular basis of the synaptome phenotypes in *Pax6*⁺ᐟ⁻ mice we will require cell-type-specific synaptome mapping methods, which are now available using conditional tagging of synaptic proteins[41]. Application of these tools will enable the identification of

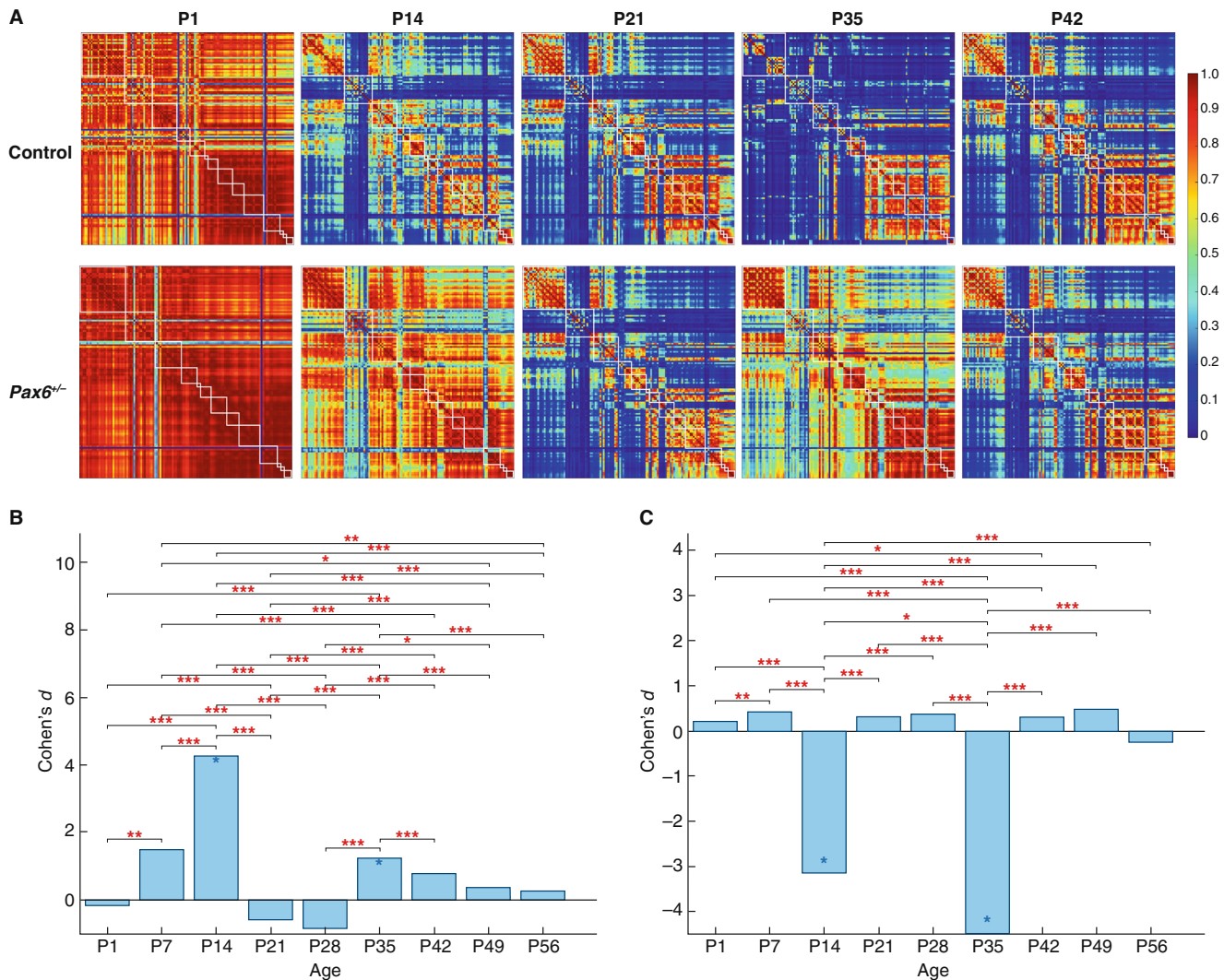

**Fig. 5 | Brain synaptome architecture is transiently disrupted in Pax6+/- mice.**
**A** At the indicated ages in control and Pax6+/- mice we compared the similarity of the synaptome between pairs of brain subregions (rows and columns) (see Fig S2 for all ages and high-resolution graphs). White boxes indicate the subregions that belong to the same main brain region (see region list in Supplementary Data 1). Scale bar, similarity values. **B** Differences (Cohen's d) in the similarity ratio between Pax6+/- and control mice for the different age groups. A significant increase in similarity ratio is seen in Pax6+/- mice at P14 and P35: blue asterisk P < 0.05, Bayesian test with Benjamini–Hochberg correction. The difference in the similarity ratio at P14 is significantly larger than at other ages: red asterisks, *P < 0.05, **P < 0.01,

***P < 0.001, two-way ANOVA with post-hoc multiple comparison test. Source data are provided as a Source Data file. See Methods for sample numbers. **C** Differences (Cohen's d) in the average small worldness between Pax6+/- and control mice for the different age groups. A significant decrease of small worldness in Pax6+/- mice is seen at P14 and P35: blue asterisk P < 0.05, Bayesian test with Benjamini–Hochberg correction. The difference in small worldness at P14 is significantly larger than at other ages: red asterisks, *P < 0.05, **P < 0.01, ***P < 0.001, two-way ANOVA with post-hoc multiple comparison test. Source data are provided as a Source Data file. See Methods for sample numbers.

synaptome phenotypes in the many types and subtypes of excitatory and inhibitory neurons. Autism and other neurodevelopmental disorders show a sex bias and it will be important to establish whether synaptome phenotypes are sex specific. Expanding the Pax6 Developmental Synaptome Atlas through the addition of further synaptic proteins (including presynaptic and inhibitory) will enhance its discovery potential, as will the development of parallel atlas resources for other germline mutations of significant health impact. Indeed, the synaptome mapping approach employed in this study can be applied to any genetic model of neurodevelopmental disease, with the potential to uncover and map through time the commonalities and distinctions that may inform underlying mechanisms and brain regional impacts. The application of synaptome mapping to human brain tissue, employing an immunological detection regime[74], adds a further layer to the discovery potential of this technology.

# Methods
## Animals
Animal procedures were performed in accordance with UK Home Office regulations and approved by Edinburgh University Director of Biological Services. Generation and characterization of Psd95eGFP/eGFP;Sap102mKO2/mKO2 knock-in mice were described previously using C57BL/6J mice[3]. Pax6+/- mice[22] were crossed with PSD95-eGFP and SAP102-mKO2 mice to generate cohorts of Pax6+/-;Psd95eGFP/eGFP;Sap102mKO2/mKO2 and control Pax6+/+;Psd95eGFP/eGFP;Sap102mKO2/mKO2 mice. We estimated the sample size according to our previous studies in mutant and wild-type mice[2,3] using a t-test analysis. Both control (c) and mutant (m) mice from both sexes were collected at nine postnatal time points: one (P1, c = 11, m = 6), seven (P7, c = 7, m = 7), fourteen (P14, c = 6, m = 7), twenty-one (P21, c = 7, m = 8), twenty-eight (P28, c = 8, m = 7), thirty-five (P35, c = 11, m = 6), forty-two (P42, c = 7, m = 6), forty-nine (P49, c = 6, m = 6) and fifty-six (P56, c = 16, m = 9) days. We only excluded

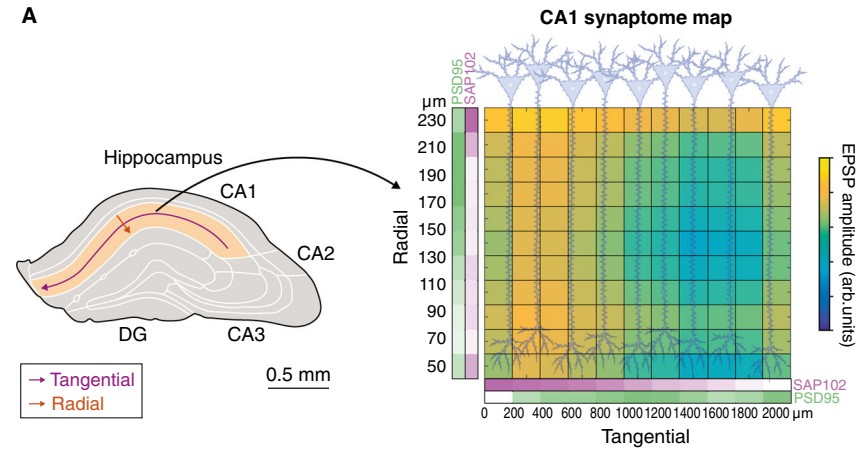

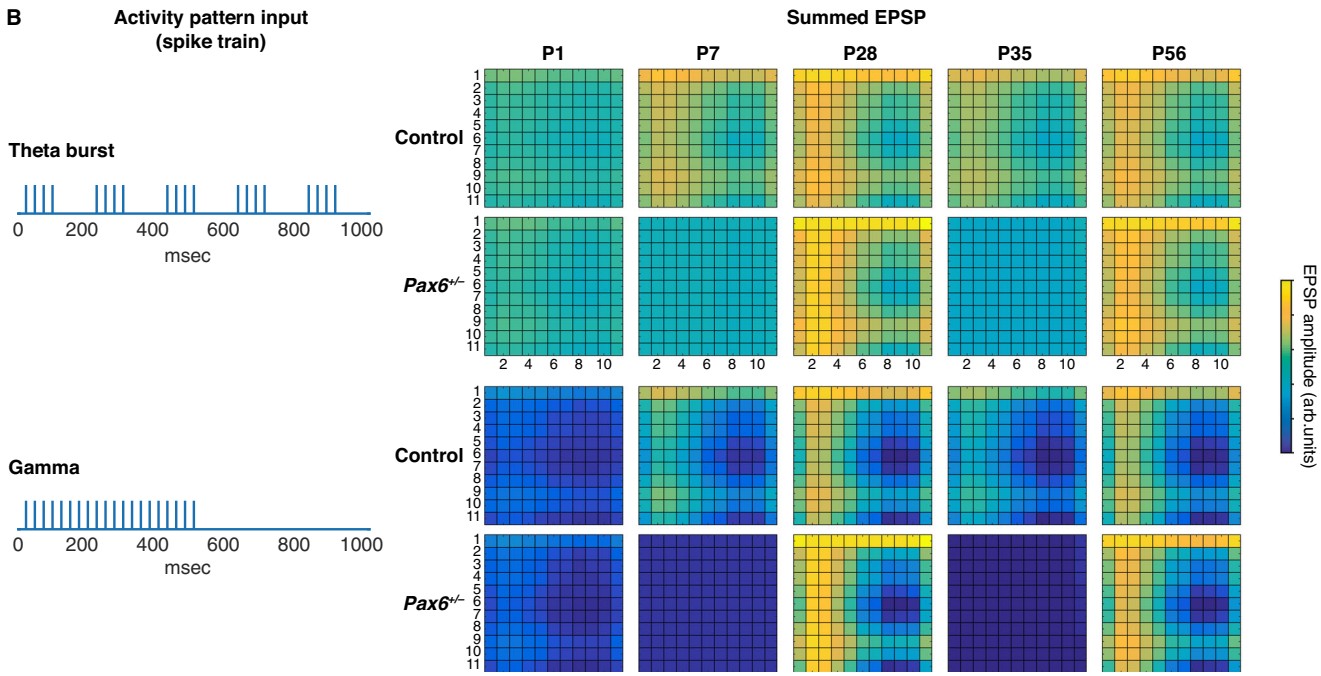

**Fig. 6 | Altered synaptic responses to physiological spike patterns in the *Pax6* mutant mouse. A** Synaptic PSD95 and SAP102 amounts (intensity) along the radial and tangential directions of the CA1 stratum radiatum, with intensity symbolized by color intensity; PSD95 (green) and SAP102 (magenta) were used as in previous work[2,3] to set synaptic properties of the computational model. Synaptic EPSP amplitudes of age groups (P1–P56) and genotype (control, *Pax6*+/−) were scaled based on PSD95 and SAP102 intensity (Supplementary Data 14 and 15). **B** Synapses were activated by spike patterns representing theta burst (top) and gamma frequency (bottom) activity. Summed EPSP response amplitudes were quantified (color bar, arbitrary units) and statistical differences between synaptic responses of control (upper) and *Pax6*+/− (lower) were assessed.

animals with absent or damaged brain regions due to errors in dissection. After imaging we only excluded images that were out of focus. No data were excluded after computational replication. Samples were allocated according to their postnatal day of collection into 9 groups (P1-P56). When collected each sample received a randomized ID number, masking their genotypes. The masking ID number was maintained during tissue and imaging processing.

### Tissue collection and sectioning
Mice were anesthetized by an intraperitoneal injection of 20% (w/v) sodium pentobarbital (Pentoject; Animalcare, York, UK): 0.01 ml for P1-P7, 0.05 ml for P14-P21, 0.1 ml for P28-P56. After complete anesthesia, phosphate-buffered saline (PBS; Oxoid, Basingstoke, UK) at 5 ml for P1-P14 and 10 ml for P21-P56 was perfused transcardially, followed by 4% (v/v) paraformaldehyde (PFA; Alfa Aesar, Heysham, UK) at 5 ml

for P1-P14 and 10 ml for P21-P56. Whole brains were dissected out and immediately postfixed at 4 °C in 4% PFA (2 h for P1-P14, 4 h for P21-P56) before transfer into 30% (w/v) sucrose (VWR Chemicals, Lutterworth, UK) at 4 °C in 1×PBS. Brains were then embedded into Optimal Cutting Temperature (OCT; CellPath, Newtown, UK) medium within a cryomould and frozen in isopentane cooled with liquid nitrogen. Brains were then sectioned in the parasagittal plane at 18 µm thickness using an NX70 cryostat (Thermo Fisher Scientific, Gloucester, UK). Cryosections were mounted on Superfrost Plus glass slides (Thermo Fisher Scientific) and stored at −80 °C.

### Tissue preparation
Parasagittal sections from left hemisphere (corresponding to sections 12–13/24 from sagittal Allen Brain Reference Atlas)[75] were washed for 5 min in PBS, incubated for 15 min in 1 mg/ml DAPI (Sigma-Aldrich,

Gillingham, UK), washed with PBS and mounted using home-made MOWIOL (Calbiochem, Nottingham, UK) containing 2.5% anti-fading agent DABCO (Sigma-Aldrich), covered with a coverslip (thickness #1.5, VWR International) and imaged the following day.

## Spinning-disk confocal microscopy

Fast high-resolution imaging was achieved using an Andor Revolution XDi system (Andor, Oslo, Norway) equipped with a UPlanSAPO 100x oil-immersion lens NA 1.4 (Olympus, London, UK), a CSU-X1 spinning-disk (Yokogawa, Runcorn, UK) and an Andor iXon Ultra monochrome back-illuminated EMCCD camera, a 2x post-magnification lens and a Borealis Perfect Illumination Delivery™ system. Images acquired with that system have a pixel dimension of 84 × 84 nm and a depth of 16 bits. A single mosaic grid was used to cover each entire brain section with an adaptive $z$ focus set up by the user to follow the unevenness of the tissue using the Andor iQ2 software. In both systems, eGFP was excited using a 488 nm laser and mKO2 with a 561 nm laser. Acquisition parameters were optimized at adult stages when the synapse intensity was high.

## Cohen's $d$ formula

Cohen's $d$ values in Fig. 2 measure the effect size of synaptome parameter changes between control and $Pax6^{+/-}$ mice as follows:

$$d = \frac{x_1' - x_2'}{s} \qquad (1)$$

where $x_1'$ and $x_2'$ are the sample average synaptome parameter for the $Pax6^{+/-}$ and control groups, respectively, for a given subregion, and $s$ is pooled standard deviation, as follows:

$$s = \sqrt{\frac{(n_1 - 1)\, s_1^2 + (n_2 - 1)\, s_2^2}{n_1 + n_2 - 2}} \qquad (2)$$

where $s_1$ and $s_2$ are the sample standard deviations in $Pax6^{+/-}$ and control groups, respectively, and $n_1$ and $n_2$ are the sample numbers of mutant and control groups, respectively.

The Cohen's $d$ values in Figs. 4 and 5 were calculated based on the Bayesian estimation by firstly inferring a probability distribution of Cohen's $d$ values. The final Cohen's $d$ value was then given as the mode of the distribution. Details of the calculation are provided in the next section.

## Bayesian analysis

Bayesian estimation[76] as used previously[3] was also applied to test the significance of the mutant effects on synaptome maps, including subtype density (Fig. 2), similarity matrix (Fig. 5), and synapse diversity (Fig. 4A), and also the difference in mutant effects on SPL versus LPL synapse subtypes (Fig. 4C). The results were finally corrected over all subregions using the Benjamini-Hochberg procedure.

Bayesian estimation was also used in calculating the Cohen's $d$ values between $Pax6^{+/-}$ and control mice (Figs. 5B, C and 4A, C). Using the Monte Carlo simulation, the sample number was upscaled based on the sample values and a t-distribution model to infer the probability density distributions/functions (PDFs) of the mean and standard deviation of the $Pax6^{+/-}$ and control groups. Then the distributions (PDFs) of Cohen's $d$ values were calculated based on the mean and standard deviation. The final Cohen's $d$ was given as the mode of its PDF, namely the Cohen's $d$ value that gave the highest probability density.

In Fig. 4C, we tested whether the SPL subtypes have a larger mutant effect size than the LPL subtypes in each subregion across all age groups. This was achieved by comparing the mutant effect size values (Cohen's $d_{mutant}$) of SPL and LPL subtypes using Bayesian analysis. We first inferred the PDFs of mutant effect size, $f_{SPL}(d_{mutant})$ for SPL and $f_{LPL}(d_{mutant})$ for LPL, respectively, with a Monte Carlo simulation. The two PDFs $f_{SPL}(d_{mutant})$ and $f_{LPL}(d_{mutant})$ were then compared using Bayesian analysis to test whether the mutant effect size in SPL was significantly larger or smaller than that in LPL, and by how much using Bayesian estimation of the Cohen's $d_{SPL-LPL}$,

$$d_{SPL-LPL} = \frac{E[f_{SPL}(d_{mutant})] - E[f_{LPL}(d_{mutant})]}{\sqrt{(s^2[f_{SPL}(d_{mutant})] + s^2[f_{LPL}(d_{mutant})])/2}} \qquad (3)$$

where $E[f]$ and $s[f]$ are and are, respectively, the expectation and standard deviation of a PDF $f$. The Cohen's $d_{SPL-LPL}$ of SPL against LPL and the corresponding significance $P$ values were calculated for each of the 131 subregions in all age groups. The Benjamini-Hochberg multiple comparison correction was finally applied over all subregions to generate the adjusted $P$ values.

## Similarity matrices and network analysis

Each row/column in the matrix represents one delineated brain sub-region at one age of either control or mutant group (Figs. 5A and S2). Elements in the matrix are the synaptome similarities between two subregions quantified by differences in standardized synaptome parameters. The similarity ratio ($S_{ratio}$) of the matrix in the $Pax6^{+/-}$ and control mice of different ages in Fig S2 is calculated as the similarity of two subregions from different main regions (corresponding to areas outside the white boxes distributed diagonally in the similarity matrices in Figs. 5A and S2) divided by the similarity of two subregions in the same main region (the areas marked by the white boxes lying on the diagonal in Figs. 5A and S2). A high $S_{ratio}$ value indicates a homogeneous synaptome similarity distributed over all columns/rows in the matrix, whereas a low ratio indicates homogeneous synaptome similarity only within the same main region. Details concerning the calculation of similarity matrix and ratio have been published previously[2,3].

After the $S_{ratio}$ is calculated for each subregion in the $Pax6^{+/-}$ and control mice of different ages, the Bayesian estimation was used to provide a developmental trajectory of Cohen's $d$ with $P$ values in Fig. 5B.

The network analysis was based on the similarity matrices of individual brain sections quantified in a similar way to those in Figs. 5A and S2. Nodes in the network are representations of the delineated subregion. The small worldness is the topology quantification of the network, where the whole set of nodes is divided into small and clustered groups: nodes within the same groups are highly connected/similar, whereas those between groups are disconnected/dissimilar[77]. Details of the small-worldness calculation can be found in our previous studies[2,3]. With the small-worldness values calculated for each age in the $Pax6^{+/-}$ and control mice, the Bayesian test was used to calculate the Cohen's $d$ and $P$ values in Fig. 5C.

## Computational modeling of synaptic responses

Computational modeling of synaptic responses was based on our previously described model[3] representing physiology in the CA1 stratum radiatum, briefly outlined below. Here we extend this model to include the effects of genotype and age. These models simulate changes in synapse physiology, EPSP amplitudes, short-term plasticity and temporal summation based on observations from neurons where PSD95 and SAP102 expression is altered[33,44–47].

**Synaptic scaling representing genotype and age.** The amount of PSD95 and SAP102 in synapses along the radial and tangential directions of the CA1 stratum radiatum (Figs. 6 and S4) is derived from the fluorescence intensity measurement of individual synaptic puncta and represented by color intensity; PSD95 (green) and SAP102 (magenta), as described in previous work[2,3], were used to scale the synaptic properties of the computational model to represent differences in

genotype and age. To model synaptic physiology corresponding to P1, P7, P28, P35 and P56 in control and $Pax6^{+/-}$ animals, differences in the intensity of PSD95 and SAP102 along radial as well as tangential directions of CA1sr were computed as outlined below.

The CA1 stratum radiatum was delineated into four radial and ten tangential subregions, for each of the genotypes and age groups. Next, we computed the geometric mean over individuals (control $N = 5$, $Pax6^{+/-}$ $N = 4$) of intensity of PSD95 and SAP102. Then, for each radial or tangential direction and protein, we normalized data according to the following. We computed the mean for each age point and genotype (the mean expression level along a direction for that age and genotype) and from this identified the minimum and the maximum value over all ages and genotypes. Normalization was done by subtracting the minimum value and dividing by the span (the difference between the maximum value and the minimum value). Thus, for radial and tangential values of PSD95 and SAP102, the intensity of expression was compared over all time points regardless of genotype. This relative intensity was used to scale the spatial distribution of synaptic puncta used in previous work[3]. This scaling thus takes into consideration the observation that intensities of PSD95 and SAP102 increase from P1 and attain a maximum at 3 months[2], which is the age of the mice in the study by Zhu et al.[3], and allows for a comparison of $Pax6^{+/-}$ versus control relative intensity levels over the age points studied.

In the model, weighted spatial distribution of synaptic puncta described above were used to scale the amplitude of short-term depression and facilitation (the PSD95 and SAP102 tangential gradient decrement factor, respectively), as well as the time constant of short-term depression and facilitation (the PSD95 and SAP102 radial decrement factor, respectively). The model is constrained to consider only age-dependent and genotype-dependent changes in synapse protein composition and does not consider potential changes in dendritic morphology or other neuronal properties. Synaptic responses following a stimulation pattern were first quantified for each synapse as the sum of synaptic max amplitudes reached following each of the 20 stimulus pulses. Differences in synaptic responses between control and $Pax6^{+/-}$ mice of all age groups and all stimulation patterns were assessed using a paired $t$-test with corrections for multiple comparisons using Benjamini-Hochberg correction, where all the 121 synaptic responses of the control case were compared with those from the $Pax6^{+/-}$ case.

## Computational model of spatial differences in PSD95 and SAP102

The following text is adapted from Zhu et al.[3] where the model was first described and models synaptome data from the CA1 stratum radiatum of mice aged 3 months.

**Synaptic responses.** Synaptic EPSPs evoked by an incoming event (transmitter release following a presynaptic spike) were described by a bi-exponential function. Parameters τ1 and τ2 were set to reproduce a fast ionotropic synaptic AMPA-type time course.

$$V = A_e \times (exp(-t/\tau1) - exp(-t/\tau2)) \tag{4}$$

where $A_e = \Pi_i 1 \times A_{tfi} \times A_{tdi}$, τ1 = 3.0 ms, τ2 = 0.4 ms, i index of all preceding spikes.

Short-term synaptic changes followed a formalism described by Tsodyks and Markram[78] and Varela et al.[79]. We included one fast and one slow facilitating component and one depressing component, all of which affected synaptic responses following the triggering one. In all figures, amplitudes are shown normalized to the amplitude of the first response.

**Depression model.**

$$A_{di} = A_d \times exp(-\Delta t/\tau_d) \tag{5}$$

where

$\Delta t_i$ is the time between the preceding event i and the present event

$A_d = A_{d0} \times S_{Ad}$, $S_{Ad}$ is normalized tangential PSD95 size factor*, [0, 1]

$\tau_d = \tau_{d0} \times S_{Td}$, $S_{Td}$ is normalized radial PSD95 size factor*, [0, 1]

$A_{tdi} = \max(\Sigma_i(1 - A_{di}), 0)$, total depressing response was limited to positive values.

**Fast facilitation, F1.**

$$A_{fi} = A_f \times exp(-\Delta t_i/\tau_f) \tag{6}$$

where

$A_f = A_{f0} \times S_{Af}$, $S_{Af}$ is normalized tangential SAP102 size factor*, [0, 1]

$\tau_f = \tau_{f0} \times S_{Tf}$, $S_{Tf}$ is normalized radial SAP102 size factor*, [0, 1]

**Slow facilitation, F2.**

$$A_{si} = A_s \times exp(-\Delta t_i/\tau_s) \tag{7}$$

where

$$A_s = A_{s0}$$

$$\tau_s = \tau_{s0}$$

The total facilitatory response had a saturation at 3.3 times the unit response[80].

$$A_{tfi} = \min(1 + \Sigma_i(A_{fi} + A_{si}), 3.3) \tag{8}$$

*The experimental tangential and radial profile data of PSD95 and SAP102 normalized size data[3] were used to set the differential model parameter values along the spatial dimension.

**Estimation of free model parameters from data.** Free model parameters were set to replicate experimental data of synaptic amplitudes in response to a 10 cycle theta-burst protocol[47]. The model was fitted to amplitude data from theta-burst experiments for bursts 1, 2, 8 and 10 in a 10 burst protocol. Verification tests showed that including a second, potentially slower, depression factor did not significantly reduce the fitting error. Furthermore, PSD95-related parameters were set to replicate the respective paired-pulse facilitation fractional differences between recordings in tissues from wild-type and knockout animals (IPI = 25, 50, 100, 200 ms)[44]. For the estimation of the parameters in knockout models, only $A_{d0}$, $\tau_{d0}$ and $A_{f0}$ were allowed to change. Verification and parameter sensitivity tests showed that inclusion of the three other parameters did not significantly affect the fitting error. Simulations were performed using MATLAB, R2020a with a time discretization of 1 ms. Time constants τ in unit ms and amplitudes in a.u (arbitrary units).

## Reporting summary

Further information on research design is available in the Nature Portfolio Reporting Summary linked to this article.

## Data availability

The synaptic data generated in this study have been deposited at Edinburgh DataShare[42] and project website[43]. Raw imaging data can be

made available on request. There are no restrictions on data availability. Source data are provided with this paper.

## Code availability

Analysis scripts used in this manuscript are available on GitHub (https://github.com/rickqiu1981/Mouse-synaptome-Pax6-disease).

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

## Acknowledgements

C. McLaughlin, E. Sigfridsson, B. Notman, R. Dahan, G. Varga, B. Koniaris, A. Bujalance and N. Skene for advice and technical assistance. C. Davey for editing. D. Maizels for artwork. EMBO Long-Term Fellowship (ALTF 1176-2015) L.T.R. Simons Foundation Autism Research Initiative (529085) L.T.R., N.H.K., S.G.N.G. The European Research Council (ERC) under the European Union's Horizon 2020 Research and Innovation Programme (695568 SYNNOVATE) Z.Q., E.B., S.G.N.G.. Wellcome Technology Development Grant (202932/Z/16/Z) R.G., Z.Q., S.G.N.G. For the purpose of open access, the author has applied a CC-BY public copyright licence to any Author Accepted Manuscript version arising from this submission.

## Author contributions

L.T.R. planned the experiments, collected, imaged and analyzed brain samples, performed image and data analyses. Z.Q. developed software and performed image and data analyses. R.G. performed data handling and built the website. E.F. analyzed data and performed computational modeling of neural activity. E.B. provided data on synapse protein turnover. N.H.K. and D.J.P. provided supervision. S.G.N.G. supervised the project and wrote the manuscript.

## Competing interests

The authors declare no competing interests.
