## [Peer Review File · Nature Communications]

Developmental disruption and restoration of brain synaptome architecture in the murine Pax6 neurodevelopmental disease modelREVIEWER COMMENTS

Reviewer #1 (Remarks to the Author):

The group has previously established a pipeline for "synaptome mapping" and, in the present study, applied the technology to apply Pax6 heterozygous mutant mice, a model for neurodevelopmental disorders. The strategy utilizes fluorescent protein-tagged knock-in mice, i.e., PSD95eGFP/eGFP and SAP102mKO2/mKO2, as well as spinning disk-microscopy to evaluate postsynaptic protein complex at the quantitative level throughout the brain, so that ultimately produce "synaptome atlas" from the newborn stage to 8 weeks of age. Although the total number of synapses was not different between the wild type and Pax6 mutant mice, there were striking changes in the regional composition of excitatory synapse types and subtypes. In addition, these synaptome phenotypes emerged transiently in two age-windows. It is of note that the synaptic phenotypes were reversed within 2-3 weeks of onset. Not only the phenotypes are interesting showing diversity of synapses, but also the technology is sophisticated and will be useful for researchers in the related fields. Thus, the work will attract the eyes of many readers of Nature Communications. This reviewer is quite positive for the acceptance of this manuscript, yet the following critical issues should be addressed:

Major comments:

- 1) Regarding the strategy, the sex of mice should not be mixed when Pax6 mutant samples were used because there are differences in brain volume decrease between male and female Pax6 mutant rats; female brains are more affected by Pax6 haploinsufficiency (Hiraoka et al., PLOS One, 2016). It is possible that PAX6 may be related with severe autism spectrum disorder often shown in girls, considering the previous genetic data that autism subjects containing both males and females show a high LOD score at the chromosome 11.13 where PAX6 gene is located, while that the subjects containing only males do not show a significance in the locus (Autism Genome Project Consortium, Nat Genet, 2007). It would be ideal to reanalyze the synaptome data by separating the two sexes.
- 2) Although the synaptome phenotypes are intriguing, their underlying mechanisms and implications are not discussed very well. Pax6 is expressed from the beginning of the neural development in neural stem/progenitor cells in the entire neural tube (except the midbrain as mentioned above), but disappear in most of the neurons except those in certain brain regions such as the olfactory bulb, amygdala, thalamus, cerebellum etc. (Kikkawa et al., Brain Res, 2019). Why there are phenotypes, for example, in cortical neurons, in which Pax6 is not expressed? Are the phenotypes severe in the diencephalic or cerebellar neurons where Pax6 is expressed? Why at P7 the most drastic change occurs? Why Type 2 (expressing only SAP) showed more drastic changes at P7 than Type 1 (expressing only PSD95)? Why there were two waves of the phenotypes during postnatal 8 weeks even though the expression of Pax6 might not be waved during the period? If there are the two waves, what does it mean regarding phenotypes of Pax6 mutant mice at the behavioral/physiological level? PAX6/Pax6 gene is originally identified as a responsible gene for aniridia in human and mouse (Ton et al., 1991; Hill et al., 1991), and the eye phenotypes are reported to be diverse in Pax6 mutant mice (Kanakubo et al., 2006). Are there any relations between the eye phenotypes and the synaptome in the occipital cortex?
- 3) In Fig. 6, the authors suddenly jump into simulation in electrophysiology of the hippocampus. It would be better to show the synaptome atlas in the hippocampus as a supplemental figure before moving to this part. There is an interesting report describing neuronal transmission in DG, CA3, and CA1 (Cizeron et al., 2004). Pax6 is expressed in the adult rodent hippocampus (Maekawa et al., 2005; and other literatures), where neurogenesis occurs. Could the authors discuss on their data in regard with the decreased neurogenesis in the Pax6 mutant mouse?
- 4) It would be interesting to investigate phenotypes in inhibitory neurons in Pax6 mutant because the gene is expressed in a part of the ganglionic eminence producing those population of neurons. In some model animals of autism spectrum disorder, imbalance between activities of excitatory and inhibitory neurons is considered to be

critical for the phenotypes. Moreover, it would be of useful if the authors could analyze the synaptome of Pax6 mutant mice at later stages (LSA3 or later) because hippocampus neurogenesis is decreased in Pax6 $-/+$ rats (Maekawa et al., 2005) and Pax6 expression is decreased with age in the wild type mice (Tripathi & Mishra, 2010; Srivastava et al., 2018).

Minor comments:

- 1) In Introduction, Pax6 is NOT expressed in the midbrain (see Osumi et al., Stem Cells, 2008, for example), and thus the sentence "Pax6 is expressed during embryogenesis in progenitor cells giving rise to forebrain, midbrain and hindbrain structures, and postnatally in subsets of diencephalic neurons" should be rewritten. Hiraoka et al. (2016) used spontaneous Pax6 mutant RATS, not mice. The authors should also refer a good review article by Manuel et al. (Front. Cell. Neurosci, 2015) including one of the co-authors.
- 2) Explanation for stages of life synaptome architecture (LSA) should appear in Introduction.
- 3) There are several types of Pax6 mutant mice, and the authors should refer Hill et al. (1991) and/or Roberts (1967) when they mention the mice used for the first time.
- 5) It would be better to explain the character of PSD95 and SAP102 in Introduction for general readers of Nature Communications. In addition, classification of 37 synapse subtypes and their relations with Type 1-3 are not easy to understand. It would be helpful to make a table or graphical summary.
- 6) In p5, line 6, Week five (P35-P42) should read Week six (P35-42).
- 7) Legend for Fig. 2 should include explanation of Cohen's d score shown as a red-blue bar.
- 8) Regarding p5, line 30-, the legend is not easy to read. How they calculate the score of Y-axis? Moreover, how short protein lifetime and long protein lifetime (LPL) is distinguished?
- 9) There is a gap in the stages shown in Fig. 5A and Fig. 6B, which should be consistent. Matrix heatmap is needed regarding Fig. 6.

Reviewer #2 (Remarks to the Author):

This study on synaptome mapping in Pax6 mutant mice reveals two waves of synaptome changes that are temporally deviated from normal developmental programs but eventually normalized by compensatory changes. This study uses two tagged excitatory synaptic marker proteins (PSD-95 and SAP102) and analyzes almost a trillion PSD-95/SAP102-containing synapses. They find that the synapses with rapid protein turnover (SPL synapses) are most strongly affected and show resilient recovery. These changes seem to occur in various brain regions, suggesting that age-dependent recovery of synaptome compositions is widespread. In addition, this study finds two critical developmental periods during which synaptome recovery strongly occurs: the weaning stage becoming independent from mothers, and the stage of sexual maturation and adult behaviors. These results suggest that, in some mouse models of neurodevelopmental disorders, spontaneous recovery of synaptomes occurs in two distinct waves in the case of Pax6 mutant mice, and provides molecular/synaptic explanations on why genetic/environmental impacts are rescued spontaneously over the courses of postnatal brain development. I have only the following minor comments.

1. It is unclear why the authors selected PSD-95 and SAP102 among the four known members of the PSD-95 family of synaptic scaffolding proteins. This should be explained.
2. The stages of decreased SAP102-containing synapses seem to be correlated with those of reduced EPSP amplitudes. I am wondering whether these two findings could be mechanistically correlated.

3. Are there any brain regions that are more strongly affected in synaptome compositions than the other regions? This should be described and explained for possible reasons.

4. What would be the potential mechanisms that lead to the changes in the composition of PSD-95/SAP102-containing synapses in Pax6 mice. Is it known whether Pax, a transcriptional factor, directly or indirectly regulates the expression PSD-95/SAP102 or any other synaptic proteins? Do you expect them to be different for the two time windows/waves?

5. What might be the potential mechanisms that spontaneously normalize synaptome compositions? Do you expect them to be different for the two time windows/waves?

6. Spatial EPSPs were measured in the hippocampus. Was the hippocampus the brain region with the strongest synaptome deviation? The results in Figure 2 do not clearly say which brain regions are most strongly affected, although the cortex seems to be the case.

7. The following sentence seems to be incorrect: "but during the second and third postnatal weeks (P7-P21) strong synaptome phenotypes emerged in most brain regions,"

Reviewer #3 (Remarks to the Author):

The authors report and analyse a huge data set on the expression of synaptic proteins PSD95 and SAP102. These data will be of considerable interest to the whole neuroscience community. The methodology is complicated due to the many types of analyses performed, and the authors have done a good job explaining it - however, despite many readings, I failed to understand certain parts. Due to short reviewing time and since I was suggested by the editor to focus on the modelling, I've only looked at the computational part in detail.

1) My main concern is that the authors adjust the depression and facilitation parameters based on PSD95 and SAP102 data. This was done already in Zhu et al. 2018, but it was never argued for. What is the rationale behind this?

The classical understanding behind short-term plasticity in the computational neuroscience field is that it represents the availability of presynaptic neurotransmitter vesicles. It has also been hypothesized that, postsynaptically, receptor desensitization could contribute to short-term depression. However, the article Varela et al. which the authors refer to in their previous paper Zhu et al. 2018 shows that post-synaptic manipulations do not affect the short-term depression and that it has a presynaptic origin. And even if desensitization of neurotransmitter receptors was the driving phenomenon behind STP, how would the amount of SAP102 and PSD95 affect it? All in all, I cannot imagine how the postsynaptic proteins SAP102 and PSD95, both of which seem to be related to anchoring of glutamatergic receptors to the post-synaptic spine membrane, could affect the short-term depression/facilitation parameters (I know these proteins are related to long-term plasticity though). I could also understand if just the total synaptic strength was adjusted based on e.g. the relative expression of these two proteins.

Minor comments:

2) The authors write "Synaptic gradients along the radial and tangential directions (Fig 6, S3) are represented with gradient intensity symbolized by color intensity; PSD95 (green) and SAP102 (magenta) were used as in previous work". This is not very obvious from Figure 6A. Do I understand right, that for each of the 11x11 squares (that make up the 230um x 2000um array) you have a PSD95 and SAP102 expression value, and from

those data you average and get the 1D colour maps below and on the left side of the array? If so, I would suggest that you plot the 11x11 green and magenta data in the picture instead of the orange patches, for example so that in each of the 11x11 squares the upper left part of the square shows the PSD95 expression (white to green) and the lower left part shows the SAP102 expression (white to magenta), and then be more explicit about this in the figure caption.

2b) But if for each of the 11x11 arrays you only save four gradients (PSD95 radial+tangential, SAP102 radial+tangential), how then do you get again 11x11 models in Figure 6B? Or are these based on the raw 11x11 data? (Each square seems to report the data from a single simulation.)

3) The authors write: "Short protein lifetime (SPL) (subtypes 6, 8, 11, 29, 28, 31) and long protein lifetime (LPL) (subtypes 5, 2, 34, 3, 35, 5, 20) synapses had skewed distributions, with SPL synapses disrupted in more brain regions than LPL synapses.". How was the lifetime measured, and how were the subtypes categorized into SPL and LPL? I thought the subtypes were assigned based on static data, not data including different time instants. (Let me know if this was described in the text and I missed it).

4) Table S14 has a cryptic cell "Removing 157" in the P56 data.

Response to reviewers

The reviewers' comments are in italics and the authors' responses are in normal font.

Reviewer #1:

The group has previously established a pipeline for “synaptome mapping” and, in the present study, applied the technology to apply Pax6 heterozygous mutant mice, a model for neurodevelopmental disorders. The strategy utilizes fluorescent protein-tagged knock-in mice, i.e., PSD95eGFP/eGFP and SAP102mKO2/mKO2, as well as spinning disk-microscopy to evaluate postsynaptic protein complex at the quantitative level throughout the brain, so that ultimately produce “synaptome atlas” from the newborn stage to 8 weeks of age. Although the total number of synapses was not different between the wild type and Pax6 mutant mice, there were striking changes in the regional composition of excitatory synapse types and subtypes. In addition, these synaptome phenotypes emerged transiently in two age-windows. It is of note that the synaptic phenotypes were reversed within 2-3 weeks of onset. Not only the phenotypes are interesting showing diversity of synapses, but also the technology is sophisticated and will be useful for researchers in the related fields. Thus, the work will attract the eyes of many readers of Nature Communications. This reviewer is quite positive for the acceptance of this manuscript, yet the following critical issues should be addressed:

We are grateful to the reviewer for their supportive comments.

Reviewer 1 point 1

Regarding the strategy, the sex of mice should not be mixed when Pax6 mutant samples were used because there are differences in brain volume decrease between male and female Pax6 mutant rats; female brains are more affected by Pax6 haploinsufficiency (Hiraoka et al., PLOS One, 2016). It is possible that PAX6 may be related with severe autism spectrum disorder often shown in girls, considering the previous genetic data that autism subjects containing both males and females show a high LOD score at the chromosome 11.13 where PAX6 gene is located, while that the subjects containing only males do not show a significance in the locus (Autism Genome Project Consortium, Nat Genet, 2007). It would be ideal to reanalyze the synaptome data by separating the two sexes.

The reviewer refers to a published study of brain volume that reports differences between adult (10 week old) male and female Pax6 rats. In our study of Pax6^{+/-} mice we did not find any synaptome phenotypes in adult mice when we combined the data from males or

females (Fig 2, S1) or when we analysed female data alone (see figure below). Unfortunately, we do not have sufficient male and female mice at every age point to perform a systematic study. In the revised discussion section “Technical Limitations and Future Perspectives” we have acknowledged that future studies could look at sex differences.

The synaptome of adult female Pax6^{+/-} mice is the same as that of female control mice. The difference (Cohen's d) in density of 37 subtypes in 131 brain subregions at nine ages from birth to P56 between female Pax6^{+/-} and female control mice. Significantly different (P<0.05, Benjamini-Hochberg corrected) subregions are shown and colored according to effect size (Cohen's d scale bar). N=14 control, N=8 Pax6^{+/-}.

Reviewer 1 point 2

Although the synaptome phenotypes are intriguing, their underlying mechanisms and implications are not discussed very well.

We agree that our discussion of the underlying mechanisms and implications of the work was limited and have accordingly expanded the discussion in several places, including adding a section entitled “Spatiotemporal synaptome phenotypes and Pax6 expression” where we address the points raised by the reviewer below:

Pax6 is expressed from the beginning of the neural development in neural stem/progenitor cells in the entire neural tube (except the midbrain as mentioned above), but disappear in most of the neurons except those in certain brain regions such as the olfactory bulb, amygdala, thalamus, cerebellum etc. (Kikkawa et al., Brain Res, 2019). Why there are phenotypes, for example, in cortical neurons, in which Pax6 is not expressed?

The new section includes the following: “It is likely, therefore, that the abnormalities in synaptome architecture reported here could arise from both the reduction in Pax6 expression at postnatal ages as well as from residual defects in neurons whose lineages no longer express Pax6. During a critical period of early development, Pax6 protects early progenitors from erroneous specification by inductive signals and thereby controls the identity of neurons long after Pax6 has ceased to be expressed⁵². Neurons that would be

expected to have cell autonomous defects in *Pax6*^{+/-} mice project their axons to neurons that do not express Pax6, such as midbrain neurons, and these postsynaptic neurons might alter their PSD95 and SAP102 expression as a result of the mutation-dependent changes in neuronal activity in the input neurons. Consistent with this, Vitalis and colleagues have shown that loss of Pax6 in populations of neurons causes cell non-autonomous phenotypes in populations of neurons whose lineages have never expressed Pax6⁴⁸.”

Are the phenotypes severe in the diencephalic or cerebellar neurons where Pax6 is expressed?

We have added the following sentence in the Discussion: “Pax6 is expressed postnatally in diencephalic and cerebellar neurons, where we observed strong phenotypes at P7 and P14 in type 2 synapses and at P35 in type 1 synapses.”

Why at P7 the most drastic change occurs? Why Type 2 (expressing only SAP) showed more drastic changes at P7 than Type 1 (expressing only PSD95)?

The most drastic changes at P7 are in type 2 and type 3 synapses, which both express SAP102. At P7, SAP102 expression increases strongly (Cizeron et al., 2020) and the interference with this normal trajectory of expression in *Pax6*^{+/-} mice can account for this observation. At P7, PSD95 is expressed at very low levels and increases at later ages.

Why there were two waves of the phenotypes during postnatal 8 weeks even though the expression of Pax6 might not be waved during the period?

In the new Discussion section “Spatiotemporal synaptome phenotypes and Pax6 expression” we now explain how the phenotype waves could arise secondary to the expression pattern of Pax6.

If there are the two waves, what does it mean regarding phenotypes of Pax6 mutant mice at the behavioral/physiological level?

In the new section in the Discussion entitled “Masking genetic phenotypes in postnatal development” we state:

“The age-windows during which synaptome architecture was restored in *Pax6*^{+/-} mice correspond to two important transitions in the life of a mouse: from dependency on the mother to independent feeding in LSA-I, and the attainment of sexual maturity and adult behaviors in LSA-II. By ensuring normal trajectories of synaptome architecture during

these crucial transitions, maladaptive behaviors caused by underlying genetic variation would be minimized and the mice more likely to survive. The brain of young animals is highly enriched in SPL synapses, providing a pool of synapse subtypes capable of rapidly remodeling and repairing the synaptome architecture during development.”

In the Discussion section entitled “Pax6 mutation transiently disrupts brain network structure and function” we present data showing that the electrophysiological outputs of the CA1 stratum radiatum (a part of the brain involved with spatial representations and memory) are transiently altered in *Pax6*^{+/-} mice.

PAX6/Pax6 gene is originally identified as a responsible gene for aniridia in human and mouse (Ton et al., 1991; Hill et al., 1991), and the eye phenotypes are reported to be diverse in Pax6 mutant mice (Kanakubo et al., 2006). Are there any relations between the eye phenotypes and the synaptome in the occipital cortex?

This is an interesting question. To address this, we asked if there were differences between the synaptome of control and *Pax6*^{+/-} mice in layers (1, 2-3, 4-5, 6) of visual areas in the occipital cortex and found significant differences at P7, P14, P21, P35 and P42. We have included this analysis in a new Table S16 and noted these changes in the results with the sentence “The strongest phenotypes were observed in the neocortex, including visual cortex (Fig 2, S1, Table S16).”

Reviewer 1 point 3

In Fig. 6, the authors suddenly jump into simulation in electrophysiology of the hippocampus. It would be better to show the synaptome atlas in the hippocampus as a supplemental figure before moving to this part. There is an interesting report describing neuronal transmission in DG, CA3, and CA1 (Cizeron et al., 2004). Pax6 is expressed in the adult rodent hippocampus (Maekawa et al., 2005; and other literatures), where neurogenesis occurs. Could the authors discuss on their data in regard with the decreased neurogenesis in the Pax6 mutant mouse?

We agree with the reviewer’s suggestion of showing the synaptome atlas data from the hippocampus as a supplemental figure and have created Figure S3 and referred to it just before the simulation.

As suggested by the reviewer we have included a discussion about neurogenesis in the Discussion section “Spatiotemporal synaptome phenotypes and Pax6 expression”.

Reviewer 1 point 4

It would be interesting to investigate phenotypes in inhibitory neurons in Pax6 mutant because the gene is expressed in a part of the ganglionic eminence producing those population of neurons. In some model animals of autism spectrum disorder, imbalance between activities of excitatory and inhibitory neurons is considered to be critical for the phenotypes. Moreover, it would be of useful if the authors could analyze the synaptome of Pax6 mutant mice at later stages (LSA3 or later) because hippocampus neurogenesis is decreased in Pax6 +/- rats (Maekawa et al., 2005) and Pax6 expression is decreased with age in the wild type mice (Tripathi & Mishra, 2010; Srivastava et al., 2018).

The reviewer raises interesting points about inhibitory neurons and neurogenesis, which we have addressed in the revised discussion. In a separate publication we have demonstrated that it is possible to use the Cre-LoxP system to label endogenous synaptic proteins in specific populations of neurons (Zhu et al., 2020) and it would therefore be possible to map synaptome changes in inhibitory neurons. A study of the synaptome in aging Pax6^{+/-} mice would indeed be interesting, especially since SPL synapses are depleted in aging mice (Bulovaite et al., 2021). However, these experiments are well outside the scope of the present study as they would take another couple of years to complete.

Reviewer 1 Minor comments

In Introduction, Pax6 is NOT expressed in the midbrain (see Osumi et al., Stem Cells, 2008, for example), and thus the sentence “Pax6 is expressed during embryogenesis in progenitor cells giving rise to forebrain, midbrain and hindbrain structures, and postnatally in subsets of diencephalic neurons” should be rewritten. Hiraoka et al. (2016) used spontaneous Pax6 mutant RATS, not mice. The authors should also refer a good review article by Manuel et al. (Front. Cell. Neurosci, 2015) including one of the co-authors.

We have corrected the sentence and included the suggested references.

1) *Explanation for stages of life synaptome architecture (LSA) should appear in Introduction.*

As requested, we have expanded the explanation of the stages of the LSA in the first paragraph of the introduction.

2) *There are several types of Pax6 mutant mice, and the authors should refer Hill et al. (1991) and/or Roberts (1967) when they mention the mice used for the first time.*

We have included these references as recommended by the reviewer.

3) *It would be better to explain the character of PSD95 and SAP102 in Introduction for general readers of Nature Communications. In addition, classification of 37 synapse subtypes and their relations with Type 1-3 are not easy to understand. It would be helpful to make a table or graphical summary.*

We agree that the relationship of types and subtypes should have been clearer and we have modified the first paragraph of the results to include the following:

“We classified the synaptic puncta into three types (type 1 express PSD95 only, type 2 express SAP102 only, and type 3 express both PSD95 and SAP102), which were classified into 37 subtypes on the basis of morphological (size and shape) features (Zhu et al., 2018) (type 1 subtypes 1-11; type 2 subtypes 12-18; type 3 subtypes 19-37).”

4) *In p5, line 6, Week five (P35-P42) should read Week six (P35-42).*

This has been corrected.

5) *Legend for Fig. 2 should include explanation of Cohen’s d score shown as a red-blue bar.*

This has been corrected.

6) *Regarding p5, line 30-, the legend is not easy to read. How they calculate the score of Y-axis? Moreover, how short protein lifetime and long protein lifetime (LPL) is distinguished?*

We have reworded the legend and hope that it is now clear. How the SPL and LPL subtypes are distinguished is described in the referenced manuscript (Bulovaite et al., 2021).

7) *There is a gap in the stages shown in Fig. 5A and Fig. 6B, which should be consistent. Matrix heatmap is needed regarding Fig. 6.*

The matrix heatmap requested for Fig. 6 can be found in supplementary Figure S2, where all stages are shown.

Reviewer #2

This study on synaptome mapping in Pax6 mutant mice reveals two waves of synaptome changes that are temporally deviated from normal developmental programs but eventually normalized by compensatory changes. This study uses two tagged excitatory synaptic marker proteins (PSD-95 and SAP102) and analyzes almost a trillion PSD-95/SAP102-containing synapses. They find that the synapses with rapid protein turnover (SPL synapses) are most strongly affected and show resilient recovery. These changes seem to occur in various brain regions, suggesting that age-dependent recovery of synaptome compositions is widespread. In addition, this study finds two critical developmental periods during which synaptome recovery strongly occurs: the weaning stage becoming independent from mothers, and the stage of sexual maturation and adult behaviors. These results suggest that, in some mouse models of neurodevelopmental disorders, spontaneous recovery of synaptomes occurs in two distinct waves in the case of Pax6 mutant mice, and provides molecular/synaptic explanations on why genetic/environmental impacts are rescued spontaneously over the courses of postnatal brain development. I have only the following minor comments.

We thank the reviewer for their supportive comments and for raising the minor comments, which we have addressed below.

Reviewer 2 point 1

It is unclear why the authors selected PSD-95 and SAP102 among the four known members of the PSD-95 family of synaptic scaffolding proteins. This should be explained.

We have modified the first paragraph of the results to read:

“PSD95 and SAP102 are two of the four paralogous members of the MAGUK family and their levels are functionally important because they physically assemble proteins controlling synaptic transmission, plasticity and neuronal excitability into macromolecular complexes (Fernandez et al., 2009; Frank and Grant, 2017; Frank et al., 2016; Husi et al., 2000), and altering their expression leads to changes in synaptic and cognitive functions (Cuthbert et al., 2007; Migaud et al., 1998). In previous studies we have characterized mice carrying engineered alleles of PSD95 and SAP102 in a wide range of biochemical, anatomical, physiological and behavioral studies (Broadhead et al., 2016; Bulovaite et al., 2021; Charlesworth et al., 2016; Cizeron et al., 2020; Cuthbert et al., 2007; Fernandez et al., 2017; Fernandez et al., 2009; Frank and Grant, 2017; Frank et al., 2016; Komiyama et al., 2002; Masch et al., 2018; Migaud et al., 1998; Nithianantharajah et al., 2013; Wegner et al., 2018; Zhu et al., 2018; Zhu et al., 2020).”

Reviewer 2 point 2

The stages of decreased SAP102-containing synapses seem to be correlated with those of reduced EPSP amplitudes. I am wondering whether these two findings could be mechanistically correlated.

It is known that SAP102 and PSD95 interact directly and indirectly with glutamate receptors, and these interactions are thought to underlie the physiological modulation of EPSPs.

Reviewer 2 point 3

Are there any brain regions that are more strongly affected in synaptome compositions than the other regions? This should be described and explained for possible reasons.

To answer this point, we have added the following sentence in the results: “The strongest phenotypes were observed in the neocortex, including visual cortex (Fig 2, S1, Table S16).” As suggested by the reviewer we have included a discussion about the mechanisms in the Discussion section “Spatiotemporal synaptome phenotypes and Pax6 expression”.

Reviewer 2 point 4

What would be the potential mechanisms that lead to the changes in the composition of PSD-95/SAP102-containing synapses in Pax6 mice. Is it known whether Pax, a transcriptional factor, directly or indirectly regulates the expression PSD-95/SAP102 or any other synaptic proteins?

We note in the introduction that “Pax6, a member of the homeodomain transcription factor family, regulates the expression of many synaptic proteins including PSD95 and SAP102”. In the Discussion section entitled “Spatiotemporal synaptome phenotypes and Pax6 expression” we state “Pax6 regulates a wide range of genes including those encoding PSD95, SAP102 and other synaptic proteins such as AMPH, NRXN3, SYNGAP1, SYNPR, SYT11 and SYT17 (Quintana-Urzainqui et al., 2018).” Whether Pax6 directly binds the transcriptional regulatory regions of these genes is unknown.

Do you expect them to be different for the two time windows/waves?

We think it is very likely that the regulation of PSD95, SAP102 and other synaptic proteins will show some differences at the two time windows because there are continuous changes in the trajectories of gene expression for synaptic proteins throughout postnatal development (Skene et al., 2017; Valor et al., 2007).

Reviewer 2 point 5

What might be the potential mechanisms that spontaneously normalize synaptome compositions? Do you expect them to be different for the two time windows/waves?

Our results indicate that the SPL synapses, which can rapidly remodel their proteomes, can account for the spontaneous normalizing of the synaptome compositions at the two time windows.

Reviewer 2 point 6

Spatial EPSPs were measured in the hippocampus. Was the hippocampus the brain region with the strongest synaptome deviation? The results in Figure 2 do not clearly say which brain regions are most strongly affected, although the cortex seems to be the case.

The reviewer is correct – the neocortex showed the strongest synaptome phenotypes and we have now stated this in the results: “The strongest phenotypes were observed in the neocortex, including visual cortex (Fig 2, S1, Table S16).”

Reviewer 2 point 7

The following sentence seems to be incorrect: "but during the second and third postnatal weeks (P7-P21) strong synaptome phenotypes emerged in most brain regions,"

We think there may be a misunderstanding since the second postnatal week is P7-14 and the third week is P14-21.

Reviewer #3

The authors report and analyse a huge data set on the expression of synaptic proteins PSD95 and SAP102. These data will be of considerable interest to the whole neuroscience community. The methodology is complicated due to the many types of analyses performed, and the authors have done a good job explaining it - however, despite many readings, I failed to understand certain parts. Due to short reviewing time and since I was suggested by the editor to focus on the modelling, I've only looked at the computational part in detail.

We thank the reviewer for their view that “*These data will be of considerable interest to the whole neuroscience community*”.

Reviewer 3 point 1

My main concern is that the authors adjust the depression and facilitation parameters based on PSD95 and SAP102 data. This was done already in Zhu et al. 2018, but it was never argued for. What is the rationale behind this?

The classical understanding behind short-term plasticity in the computational neuroscience field is that it represents the availability of presynaptic neurotransmitter vesicles. It has also been hypothesized that, postsynaptically, receptor desensitization could contribute to short-term depression. However, the article Varela et al. which the authors refer to in their previous paper Zhu et al. 2018 shows that post-synaptic manipulations do not affect the short-term depression and that it has a presynaptic origin. And even if desensitization of neurotransmitter receptors was the driving phenomenon behind STP, how would the amount of SAP102 and PSD95 affect it? All in all, I cannot imagine how the postsynaptic proteins SAP102 and PSD95, both of which seem to be related to anchoring of glutamatergic receptors to the post-synaptic spine membrane, could affect the short-term depression/facilitation parameters (I know these proteins are related to long-term plasticity though). I could also understand if just the total synaptic strength was adjusted based on e.g. the relative expression of these two proteins.

Varela et al refer to some forms of postsynaptic mechanisms, but there certainly are others. It is now well established that STP is regulated by both presynaptic and postsynaptic proteins. In our own work we have found that mutations in 20 different postsynaptic proteins alter STP (Kopanitsa et al., 2018). The majority of these physically interact with PSD95, which binds membrane-spanning proteins that in turn interact with the presynaptic proteins and the neurotransmitter release machinery (Südhof, 2008). We have built on the work of Carlisle et al (Carlisle et al., 2008) showing that EPSP magnitude and kinetics and paired-pulse facilitation are affected by PSD95 and PSD93 mutations.

For SAP102, it has been shown that it influences decay kinetics of the EPSCs (Levy et al., 2015). The effect on short-term plasticity is only indirect via its interplay with PSD95 and PSD93 (Bonnet et al., 2013; Elias et al., 2006).

Minor comments:

Reviewer 3 point 2

The authors write "Synaptic gradients along the radial and tangential directions (Fig 6, S3) are represented with gradient intensity symbolized by color intensity; PSD95 (green) and SAP102 (magenta) were used as in previous work". This is not very obvious from Figure 6A. Do I understand right, that for each of the 11x11 squares (that make up the 230um x 2000um array) you have a PSD95 and SAP102 expression value, and from those data you average and get the 1D colour maps below and on the left side of the array? If so, I would suggest that you plot the 11x11 green and magenta data in the picture instead of the orange patches, for example so that in each of the 11x11 squares the upper left part of the square shows the PSD95 expression (white to green) and the lower left part shows the SAP102 expression (white to magenta), and then be more explicit about this in the figure caption.

To avoid confusion, we have modified the legend and methods to remove the word "gradient" and instead refer directly to the quantity of PSD95 and SAP102, which is measured by protein intensity. The color intensity of the green and magenta "color bars" in Figure 6A is merely an example of the levels at a particular age and was adapted from Figure 5B of Zhu et al (2018). As the reviewer suggested, these abundances in the color bars are discretized versions (means over intervals of size 1/11) of the graded abundances.

We certainly appreciate the reviewer's suggestion that we could plot the amounts of PSD95 and SAP102 in each of the 121 squares, but to do this systematically for all 9 age points would in our view not be suitable for this main figure and we hope that the adjustments to the figure legend will be sufficient to clarify the matter.

2b) But if for each of the 11x11 arrays you only save four gradients (PSD95 radial+tangential, SAP102 radial+tangential), how then do you get again 11x11 models in Figure 6B? Or are these based on the raw 11x11 data? (Each square seems to report the data from a single simulation.)

It is correct that each square in the 11x11 array reports the data from a single simulation of one synapse and the amounts of PSD95 and SAP102 in each synapse were used to set model parameters.

Reviewer 3 point 3

The authors write: "Short protein lifetime (SPL) (subtypes 6, 8, 11, 29, 28, 31) and long protein lifetime (LPL) (subtypes 5, 2, 34, 3, 35, 5, 20) synapses had skewed distributions, with SPL synapses disrupted in more brain regions than LPL synapses.". How was the lifetime measured, and how were the subtypes categorized into SPL and LPL? I thought the subtypes were assigned based on static data, not data including different time instants. (Let me know if this was described in the text and I missed it).

All these questions are answered in a separate dedicated manuscript that we reference in the text (Bulovaite et al., 2021).

Reviewer 3 point 4

Table S14 has a cryptic cell "Removing 157" in the P56 data.

The cryptic cell has been removed.

References

Bonnet, S.A., Akad, D.S., Samaddar, T., Liu, Y., Huang, X., Dong, Y., and Schlüter, O.M. (2013). Synaptic state-dependent functional interplay between postsynaptic density-95 and synapse-associated protein 102. *Journal of Neuroscience* 33, 13398-13409.

Broadhead, M.J., Horrocks, M.H., Zhu, F., Muresan, L., Benavides-Piccione, R., DeFelipe, J., Fricker, D., Kopanitsa, M.V., Duncan, R.R., Klenerman, D., *et al.* (2016). PSD95 nanoclusters are postsynaptic building blocks in hippocampus circuits. *Sci Rep* 6, 24626.

Bulovaite, E., Qiu, Z., Kratschke, M., Zgraj, A., Fricker, D., Tuck, E.J., Gokhale, R., Jami, S., Merino-Serrais, P., Husi, E., *et al.* (2021). A brain atlas of synapse protein lifetime across the mouse lifespan. *bioRxiv* 2021.12.16.472938.

Carlisle, H.J., Fink, A.E., Grant, S.G., and O'Dell, T.J. (2008). Opposing effects of PSD-93 and PSD-95 on long-term potentiation and spike timing-dependent plasticity. *J Physiol* 586, 5885-5900.

Charlesworth, P., Morton, A., Eglen, S.J., Komiyama, N.H., and Grant, S.G. (2016). Canalization of genetic and pharmacological perturbations in developing primary neuronal activity patterns. *Neuropharmacology* 100, 47-55.

Cizeron, M., Qiu, Z., Koniaris, B., Gokhale, R., Komiyama, N.H., Fransen, E., and Grant, S.G.N. (2020). A brainwide atlas of synapses across the mouse lifespan. *Science* 369, 270-275.

Cuthbert, P.C., Stanford, L.E., Coba, M.P., Ainge, J.A., Fink, A.E., Opazo, P., Delgado, J.Y., Komiyama, N.H., O'Dell, T.J., and Grant, S.G. (2007). Synapse-associated protein 102/dlgh3 couples the NMDA receptor to specific plasticity pathways and learning strategies. *J Neurosci* 27, 2673-2682.

Elias, G.M., Funke, L., Stein, V., Grant, S.G., Brecht, D.S., and Nicoll, R.A. (2006). Synapse-specific and developmentally regulated targeting of AMPA receptors by a family of MAGUK scaffolding proteins. *Neuron* 52, 307-320.

Fernandez, E., Collins, M.O., Frank, R.A.W., Zhu, F., Kopanitsa, M.V., Nithianantharajah, J., Lempriere, S.A., Fricker, D., Elsegood, K.A., McLaughlin, C.L., *et al.* (2017). Arc Requires PSD95 for Assembly into Postsynaptic Complexes Involved with Neural Dysfunction and Intelligence. *Cell Rep* 21, 679-691.

Fernandez, E., Collins, M.O., Uren, R.T., Kopanitsa, M.V., Komiyama, N.H., Croning, M.D., Zografos, L., Armstrong, J.D., Choudhary, J.S., and Grant, S.G. (2009). Targeted tandem affinity purification of PSD-95 recovers core postsynaptic complexes and schizophrenia susceptibility proteins. *Mol Syst Biol* 5, 269.

Frank, R.A., and Grant, S.G. (2017). Supramolecular organization of NMDA receptors and the postsynaptic density. *Curr Opin Neurobiol* 45, 139-147.

Frank, R.A., Komiyama, N.H., Ryan, T.J., Zhu, F., O'Dell, T.J., and Grant, S.G. (2016). NMDA receptors are selectively partitioned into complexes and supercomplexes during synapse maturation. *Nat Commun* 7, 11264.

Husi, H., Ward, M.A., Choudhary, J.S., Blackstock, W.P., and Grant, S.G. (2000). Proteomic analysis of NMDA receptor-adhesion protein signaling complexes. *Nat Neurosci* 3, 661-669.

Komiyama, N.H., Watabe, A.M., Carlisle, H.J., Porter, K., Charlesworth, P., Monti, J., Strathdee, D.J., O'Carroll, C.M., Martin, S.J., Morris, R.G., *et al.* (2002). SynGAP regulates ERK/MAPK signaling, synaptic plasticity, and learning in the complex with postsynaptic density 95 and NMDA receptor. *J Neurosci* 22, 9721-9732.

Kopanitsa, M.V., van de Lagemaat, L.N., Afinowi, N.O., Strathdee, D.J., Strathdee, K.E., Fricker, D.G., Tuck, E.J., Elsegood, K.A., Croning, M.D., Komiyama, N.H., *et al.* (2018).

A combinatorial postsynaptic molecular mechanism converts patterns of nerve impulses into the behavioral repertoire. *bioRxiv* 500447.

Levy, J.M., Chen, X., Reese, T.S., and Nicoll, R.A. (2015). Synaptic consolidation normalizes AMPAR quantal size following MAGUK loss. *Neuron* *87*, 534-548.

Masch, J.M., Steffens, H., Fischer, J., Engelhardt, J., Hubrich, J., Keller-Findeisen, J., D'Este, E., Urban, N.T., Grant, S.G.N., Sahl, S.J., *et al.* (2018). Robust nanoscopy of a synaptic protein in living mice by organic-fluorophore labeling. *Proc Natl Acad Sci U S A* *115*, E8047-E8056.

Migaud, M., Charlesworth, P., Dempster, M., Webster, L.C., Watabe, A.M., Makhinson, M., He, Y., Ramsay, M.F., Morris, R.G., Morrison, J.H., *et al.* (1998). Enhanced long-term potentiation and impaired learning in mice with mutant postsynaptic density-95 protein. *Nature* *396*, 433-439.

Nithianantharajah, J., Komiyama, N.H., McKechnie, A., Johnstone, M., Blackwood, D.H., St Clair, D., Emes, R.D., van de Lagemaat, L.N., Saksida, L.M., Bussey, T.J., *et al.* (2013). Synaptic scaffold evolution generated components of vertebrate cognitive complexity. *Nat Neurosci* *16*, 16-24.

Quintana-Urzainqui, I., Kozić, Z., Mitra, S., Tian, T., Manuel, M., Mason, J.O., and Price, D.J. (2018). Tissue-specific actions of Pax6 on proliferation and differentiation balance in developing forebrain are Foxg1 dependent. *Iscience* *10*, 171-191.

Skene, N.G., Roy, M., and Grant, S.G. (2017). A genomic lifespan program that reorganises the young adult brain is targeted in schizophrenia. *Elife* *6*, e17915.

Südhof, T.C. (2008). Neuroligins and neurexins link synaptic function to cognitive disease. *Nature* *455*, 903-911.

Valor, L.M., Charlesworth, P., Humphreys, L., Anderson, C.N., and Grant, S.G. (2007). Network activity-independent coordinated gene expression program for synapse assembly. *Proceedings of the National Academy of Sciences* *104*, 4658-4663.

Wegner, W., Mott, A.C., Grant, S.G.N., Steffens, H., and Willig, K.I. (2018). In vivo STED microscopy visualizes PSD95 sub-structures and morphological changes over several hours in the mouse visual cortex. *Sci Rep* *8*, 219.

Zhu, F., Cizeron, M., Qiu, Z., Benavides-Piccione, R., Kopanitsa, M.V., Skene, N.G., Koniaris, B., DeFelipe, J., Fransen, E., Komiyama, N.H., *et al.* (2018). Architecture of the Mouse Brain Synaptome. *Neuron* *99*, 781-799 e710.

Zhu, F., Collins, M.O., Harmse, J., Choudhary, J.S., Grant, S.G., and Komiyama, N.H. (2020). Cell-type-specific visualisation and biochemical isolation of endogenous synaptic proteins in mice. *European Journal of Neuroscience* *51*, 793-805.

REVIEWER COMMENTS

Reviewer #1 (Remarks to the Author):

This reviewer agreed with the authors' revisions to address the issues raised by this reviewer, though it is difficult to be completely satisfied in some areas.

Reviewer #2 (Remarks to the Author):

The authors have fully addressed my review comments, and I do not have additional comments.

Reviewer #3 (Remarks to the Author):

In my first review I mentioned that I did not find it a sound approach to alter the model parameters regulating short-term plasticity (STP) based on PSD95 and SAP102 expression data, since, being post-synaptically expressed, these proteins do not seem to affect STP. The authors explained that STP is altered by many post-synaptic proteins. They mention a preprint (Kopanitsa et al. 2018) from their lab where altered paired-pulse ratio (PPR) was measured for animals where genes encoding post-synaptic proteins were mutated. The authors mention 20 such genes. In the preprint I could find 11 genes affecting PPR-10ms or PPR-50ms (Fig. 2 of Kopanitsa et al.), but many of these were genes expressed both pre- and post-synaptically (e.g. those encoding glutamate receptors and PKA anchoring proteins). More importantly, in Kopanitsa et al. a PSD95 mutant showed no effect on paired-pulse ratio although it affected many other phenotypes. As for a mechanism through which PSD95 could affect vesicle release, the authors mention Sudhof et al. 2008, which discusses the structure of adhesion-based neurexin-neuroligin junctions that comprise both pre- and post-synaptic proteins, including PSD95 on the post-synaptic side. However, in my opinion this review only suggests that an interaction between PSD95 and presynaptic vesicle release is possible. As the authors are probably the first to (although implicitly) propose that PSD95 and SAP102 affect STP, I would expect them to state out in more detail, at least on a speculative level, what mechanism lies behind this effect - or if the mechanism is not known at all, to show data that strongly supports an interaction of the proposed kind.

The strongest evidence for PSD95 affecting STP comes from Carlisle et al. 2008 mentioned by the authors, which (although focusing on post-synaptic plasticity) showed that paired-pulse facilitation was altered in PSD95 mutants. However, in Suzuki and Kamiya 2016 (PMID: 26746114), PSD95 knock-out showed no effect on paired-pulse ratio for 50-ms interval, consistent with Kopanitsa et al. As for SAP102, the paper by Levy et al. mentioned by the authors also shows a negative result concerning STP, since SAP102 knockdown did not affect paired-pulse ratio.

Taken together, I don't find the support from the literature for STP being modulated by PSD95 and SAP102 to be strong although it is clear that the shape of EPSC may be affected by them. As STP is anyway not even mentioned in the manuscript, I would advise the authors to redo their simulations by only adjusting the model parameters that are affected by these postsynaptic proteins (these could include both the EPSC amplitude and decay of the EPSCs to capture an altered AMPA/NMDA ratio) rather than adjusting parameters that affect the neural dynamics through their effects on STP.

Alternatively, if the authors argue that the observations of Carlisle et al. provide strong enough evidence for the effects of PSD95 on STP, they should anyway discuss the papers showing opposite data and acknowledge that their observations showing altered synaptic responses in the Pax6 mutant are based on an assumption of STP being altered due to altered PSD95 and SAP102 expression - this is an assumption that, I believe, may be proven wrong in the future, and thus it's good to be explicit about it.

In either case, I would advise the authors to describe the methods related to the simulations. The paper should be to some extent self-explanatory without reading the other paper (Zhu et al.).

Response to Reviewers

The reviewer's comments are reproduced in italics and the authors' responses in plain text. We thank the reviewers for their comments and appreciate the time and effort taken to review our paper.

Reviewer 1

This reviewer agreed with the authors' revisions to address the issues raised by this reviewer, though it is difficult to be completely satisfied in some areas.

Reviewer 2

The authors have fully addressed my review comments, and I do not have additional comments.

Reviewer 3

Overview

Reviewer 3 questioned whether there is sufficient literature to support the view that PSD95 plays a role in STP and, because of their doubt, they suggested that we modify the parameters in our computational simulation (Fig. 6, S3). In our point-by-point response below, we describe four papers that show PSD95 plays an essential role in STP. We apologise for not including these references among our first response to review. Consequently, and in line with the reviewer's directions, we do not require any modifications to our model. However, in order to be thorough, we examined the effects of implementing the adjustments to the model suggested by the reviewer and found that minor changes in the output resulted, but these did not impact the interpretation of the results presented in our manuscript. We hope that the reviewer's concerns are now satisfied.

Point 1

In my first review I mentioned that I did not find it a sound approach to alter the model parameters regulating short-term plasticity (STP) based on PSD95 and SAP102 expression data, since, being post-synaptically expressed, these proteins do not seem to affect STP. The authors explained that STP is altered by many post-synaptic proteins.

They mention a preprint (Kopanitsa et al. 2018) from their lab where altered paired-pulse ratio (PPR) was measured for animals where genes encoding post-synaptic proteins were mutated. The authors mention 20 such genes. In the preprint I could find 11 genes affecting PPR-10ms or PPR-50ms (Fig. 2 of Kopanitsa et al.), but many of these were genes expressed both pre- and post-synaptically (e.g. those encoding glutamate receptors and PKA anchoring proteins). More importantly, in Kopanitsa et al. a PSD95 mutant showed no effect on paired-pulse ratio although it affected many other phenotypes. As for a mechanism through which PSD95 could affect vesicle release, the authors mention Sudhof et al. 2008, which discusses the structure of adhesion-based neurexin-neuroligin junctions that comprise both pre- and post-synaptic proteins, including PSD95 on the post-synaptic side. However, in my opinion this review only suggests that an interaction between PSD95 and presynaptic vesicle release is possible.

There are many forms of STP, which are defined as activity-dependent changes in synaptic transmission that occur up to several minutes after the stimulus (Zucker and Regehr, 2002). Historically, it was thought that only presynaptic proteins could play a role in forms of STP; however, this has proven to be incorrect, with many reports showing that postsynaptic proteins alter STP too. Most notably among these are the AMPA receptors (Constals et al., 2015; Frischknecht et al., 2009; von Engelhardt, 2022) and NMDA receptors (Malenka and Nicoll, 1993). Using mice carrying gene mutations, we have confirmed that these receptors, as well as their associated proteins, result in changes in STP (Carlisle et al., 2008; Kopanitsa et al., 2018; Migaud et al., 1998). Although the mechanisms by which these proteins exert their effects on the different forms of STP remain unclear, these mechanisms are not directly relevant to the present manuscript.

We note that the synaptome changes we report in *Pax6* mutant mice are based on the expression of PSD95 and SAP102, which are known to interact with glutamate receptors and many other synaptic proteins that affect synaptic plasticity, so it should not be surprising that the mutants have phenotypes.

The reviewer refers to the Kopanitsa manuscript (Kopanitsa et al., 2018), which does indeed report “20 such genes” and the reviewer then refers to a subset of 11 that interfere with PPR. It is important to note that the other 9 genes interfere with other forms of temporal processing, including the postsynaptic responses to trains of activity over a period of 2 seconds. These differential effects of genes on forms of temporal processing nicely illustrate how perturbations in different molecules selectively interfere with different forms of temporal processing recruited with different patterns of activity.

We agree with the reviewer that some of the proteins listed in figure 2 of Kopanitsa et al are expressed in the presynaptic terminals of some synapse types and in those particular cases it is not possible to claim that they control STP by either pre- or postsynaptic mechanisms. Although this is certainly an interesting point, it is not a concern for our modelling because PSD95 and SAP102 are not presynaptic proteins and we do not include the other proteins in our model.

Point 2

As the authors are probably the first to (although implicitly) propose that PSD95 and SAP102 affect STP, I would expect them to state out in more detail, at least on a speculative level, what mechanism lies behind this effect - or if the mechanism is not known at all, to show data that strongly supports an interaction of the proposed kind.

The strongest evidence for PSD95 affecting STP comes from Carlisle et al. 2008 mentioned by the authors, which (although focusing on post-synaptic plasticity) showed that paired-pulse facilitation was altered in PSD95 mutants. However, in Suzuki and Kamiya 2016 (PMID: 26746114), PSD95 knock-out showed no effect on paired-pulse ratio for 50-ms interval, consistent with Kopanitsa et al. As for SAP102, the paper by Levy et al. mentioned by the authors also shows a negative result concerning STP, since SAP102 knockdown did not affect paired-pulse ratio. Taken together, I don't find the support from the literature for STP being modulated by PSD95 and SAP102 to be strong although it is clear that the shape of EPSC may be affected by them.

We are aware of four papers that report a role for PSD95 in STP and one paper that reports a role for SAP102, as detailed below with the relevant figures extracted. Importantly, the reviewer has compared the Suzuki and Kamiya paper (Suzuki and Kamiya, 2016), which studied mossy fibre-CA3 synapses, to the papers that we have cited and modelled that study CA3-CA1 synapses. The physiological differences between these anatomically distinct synapses are well documented and are likely to impact comparisons between them.

1. Figure 3D of Migaud et al (Migaud et al., 1998), which is the first report of PSD95 having a role in STP, shows that there are clear phenotypes in paired pulse facilitation at several time intervals. It was our oversight not to have referred to this paper in our initial response to Reviewer 3.

d, Paired-pulse facilitation of fEPSPs was measured using pairs of presynaptic fibre stimulation pulses separated by 20, 50, 100 and 200 ms. Note the larger facilitation ($P < 0.05$) seen in slices from PSD-95-mutant animals (filled symbols) compared to wild type (open symbols). Each point is the mean \pm s.e.m. (error bars are smaller than point used to plot the mean value).

- The reviewer has referred to the second report of PSD95 having STP phenotypes (Carlisle et al., 2008), which is also from the authors' group and collaborators. Carlisle et al used the same protocols as Migaud et al and reproduced the findings of that study.

c, paired-pulse facilitation is enhanced in the hippocampal CA1 region of PSD-93^{-/-} mutant mice. Pairs of presynaptic fibre stimulation pulses were delivered with interpulse intervals of 25, 50, 100, or 200 ms. Significantly greater paired-pulse facilitation is seen at interpulse intervals between 25 and 100 ms in slices from PSD-93^{-/-} mutant mice (\blacktriangle , $n = 14$ slices from 7 mice) and PSD-95 mutant mice (\bullet , $n = 13$ slices from 4 mice, $*P < 0.02$, $\#P < 0.01$ compared to wild-type littermates). The open circles show paired-pulse facilitation in slices from all wild-type mice ($n = 27$ slices from 11 mice).

- The reviewer referred to the Kopanitsa preprint (Kopanitsa et al., 2018) and noted that there was not an STP phenotype in one of the figures. Unfortunately, that

figure is incorrect, whereas the supplementary figures in the paper show that there is indeed a PPF phenotype:

WT: 200 ± 4% (17 slices from 5 animals)
Mutant: 232 ± 8% (18 slices from 5 animals); $P = 0.00083$

Furthermore, this study shows phenotypes in other forms of STP (burst phenotypes) in two lines of PSD95 mutant mice (a knockout and an SH3 domain point mutation) and found STP phenotypes in both lines.

4. Horner et al (Horner et al., 2018) observed PPF phenotypes in both homozygous and heterozygous PSD95 mutant mice.

Homozygous PSD95 mutant mice:

Heterozygous PSD95 mutant mice:

5. The reviewer refers to a paper by Suzuki and Kamiya (Suzuki and Kamiya, 2016) which reports that a mouse knockout of PSD95 does not show an STP phenotype. This paper did not study CA1 synapses, but mossy fibre-CA3 synapses, which have different synaptic physiological properties to CA1 synapses, as well as marked differences in the expression of PSD95 and SAP102. The Suzuki and Kamiya study was also extremely limited in their exploration of STP, measuring PPF at only one interval (50 ms).
6. Regarding the role of SAP102 in synaptic responses to inputs, Kopanitsa et al (Kopanitsa et al., 2018) show in figure 4 a ‘burst 10’ phenotype with Cohen’s d effect size of 0.8 (and to a lesser extent the other burst phenotypes), supporting the view that SAP102 also affects temporal processing. We used these data to construct our original model (Zhu et al., 2018). In Point 3 below, we present a modified model using parameters suggested by Reviewer 3.

Point 3

As STP is anyway not even mentioned in the manuscript, I would advise the authors to redo their simulations by only adjusting the model parameters that are affected by these postsynaptic proteins (these could include both the EPSC amplitude and decay of the EPSCs to capture an altered AMPA/NMDA ratio) rather than adjusting parameters that affect the neural dynamics through their effects on STP.

Alternatively, if the authors argue that the observations of Carlisle et al. provide strong enough evidence for the effects of PSD95 on STP, they should anyway discuss the papers showing opposite data and acknowledge that their observations showing altered synaptic responses in the Pax6 mutant are based on an assumption of STP being altered

due to altered PSD95 and SAP102 expression - this is an assumption that, I believe, may be proven wrong in the future, and thus it's good to be explicit about it.

In either case, I would advise the authors to describe the methods related to the simulations. The paper should be to some extent self-explanatory without reading the other paper (Zhu et al.).

Although we have removed any doubt about the role of PSD95 in STP, for thoroughness we test-modified our model in line with Reviewer 3's suggestions: we implemented a version of the model in which the SAP102 effects are on the AMPA/NMDA ratio, as observed in Levy et al. (Levy et al., 2015), and removed its effects on STP. The key observation is that – as with our original model – there are no significant phenotypes at P28 and P56 (figure below). Thus, the modified model shows the transience of the phenotypes whereby some of the stimulation patterns do not give rise to phenotypes in the two age windows when the synaptome is observed to be restored to normal.

Legend for the figure: Altered synaptic responses to physiological spike patterns in the *Pax6*^{+/-} mutant mouse.

Synapses were activated by spike patterns representing theta burst, gamma frequency, theta frequency and gamma burst activity (top to bottom). Synaptic amplitudes of age groups (P1-P56) and genotype (control, *Pax6*^{+/-}) were scaled based on intensity (Table S14, S15). Summed EPSP response amplitudes were quantified (color bar, arbitrary units) and statistical differences between synaptic responses of control (upper) and *Pax6*^{+/-} (lower) were assessed (paired t-test: P<0.01, red; P<0.001, black), Benjamini-Hochberg corrected). The red-boxed datasets indicate no difference between control and *Pax6*^{+/-} mice at P<0.01, Students t-test, BH corrected; The black-boxed data sets indicate no difference between control and *Pax6*^{+/-} mice at P<0.001, Students t-test, BH corrected.

As in the original model, theta burst and gamma train patterns produce an alternation between phenotype and restoration, notably with restoration to normal at P28 and P56. In the modified model, theta frequency and gamma burst patterns also result in alterations. Overall, looking at the results of both models, this illustrates how temporal processing can be affected by expression patterns of PSD95 and SAP102 and how details in the model – the specific properties of the synapse - can affect which patterns are influenced and which are not, as previously discussed in Zhu et al. (2018).

A note on changes introduced into the model

Radial and tangential expressions of PSD95 affect the STP-related depression component as in the original model (Zhu et al., 2018). Radial and tangential expressions of SAP102 were removed from the STP-related components and instead were set to affect the AMPA/NMDA ratio (Levy et al., 2015). Specifically, the fast facilitation that in the original model was affected by SAP102 was set to be affected by PSD95 in the same manner as for depression (which was left unaltered). The effect on the AMPA/NMDA ratio required the addition of an NMDA component

of the EPSP for which parameters were taken from Spruston et al (Spruston et al., 1995) including rise time constant, decay time constant and relative amplitude to the AMPA component. Moreover, the SAP102 gradient data were used to scale the amplitude of the NMDA component in a multiplicative manner (radial and tangential normalized expressions both multiplied the NMDA amplitude). Data used from Spruston et al. (1995): NMDA rise time constant 7 ms; NMDA decay time constant 200 ms; NMDA amplitude was 20% of AMPA.

As requested by the reviewer we have expanded the Methods section to assist the reader.

References

Carlisle, H.J., Fink, A.E., Grant, S.G., and O'Dell, T.J. (2008). Opposing effects of PSD-93 and PSD-95 on long-term potentiation and spike timing-dependent plasticity. *J Physiol* 586, 5885-5900.

Constals, A., Penn, A.C., Compans, B., Toulme, E., Phillipat, A., Marais, S., Retailleau, N., Hafner, A.-S., Coussen, F., and Hossy, E. (2015). Glutamate-induced AMPA receptor

desensitization increases their mobility and modulates short-term plasticity through unbinding from Stargazin. *Neuron* 85, 787-803.

Frischknecht, R., Heine, M., Perrais, D., Seidenbecher, C.I., Choquet, D., and Gundelfinger, E.D. (2009). Brain extracellular matrix affects AMPA receptor lateral mobility and short-term synaptic plasticity. *Nat Neurosci* 12, 897-904.

Horner, A.E., McLaughlin, C.L., Afinowi, N.O., Bussey, T.J., Saksida, L.M., Komiyama, N.H., Grant, S.G., and Kopanitsa, M.V. (2018). Enhanced cognition and dysregulated hippocampal synaptic physiology in mice with a heterozygous deletion of PSD-95. *European Journal of Neuroscience* 47, 164-176.

Kopanitsa, M.V., van de Lagemaat, L.N., Afinowi, N.O., Strathdee, D.J., Strathdee, K.E., Fricker, D.G., Tuck, E.J., Elsegood, K.A., Croning, M.D., Komiyama, N.H., *et al.* (2018). A combinatorial postsynaptic molecular mechanism converts patterns of nerve impulses into the behavioral repertoire. *bioRxiv* 500447; doi: <https://doi.org/10.1101/500447>

Malenka, R.C., and Nicoll, R.A. (1993). NMDA-receptor-dependent synaptic plasticity: multiple forms and mechanisms. *Trends in Neurosciences* 16, 521-527.

Migaud, M., Charlesworth, P., Dempster, M., Webster, L.C., Watabe, A.M., Makhinson, M., He, Y., Ramsay, M.F., Morris, R.G., Morrison, J.H., *et al.* (1998). Enhanced long-term potentiation and impaired learning in mice with mutant postsynaptic density-95 protein. *Nature* 396, 433-439.

Spruston, N., Jonas, P., and Sakmann, B. (1995). Dendritic glutamate receptor channels in rat hippocampal CA3 and CA1 pyramidal neurons. *The Journal of physiology* 482, 325-352.

Suzuki, E., and Kamiya, H. (2016). PSD-95 regulates synaptic kainate receptors at mouse hippocampal mossy fiber-CA3 synapses. *Neuroscience Research* 107, 14-19.

von Engelhardt, J. (2022). Role of AMPA receptor desensitization in short term depression—lessons from retinogeniculate synapses. *The Journal of Physiology* 600, 201-215.

Zucker, R.S., and Regehr, W.G. (2002). Short-term synaptic plasticity. *Annu Rev Physiol* 64, 355-405.

REVIEWERS' COMMENTS

Reviewer #3 (Remarks to the Author):

The authors have addressed some of my concerns by referring to articles where STP was shown to be at least partly mediated by post-synaptic mechanisms.

I have additional remarks on the newly added description of the computational model.

In the "Depression model" and "Fast facilitation" the authors refer to normalized "tangential" and "radial" PSD95 or SAP102 size factors. I understood from their previous reply letter that in each of the 11x11 squares the expression values of PSD95 and SAP102 were used as such instead of calculating the mean radial/tangential expression values. Does the same apply here? If so, please correct.

In the reply letter the authors stated that the mechanism of how PSD95 and SAP102 affect STP remains unclear, but it seems to me that the authors here have assumed a very specific mechanism in the computational analysis.

Namely, they seem to assume that the amplitude of depression is determined by the tangential PSD95 size factor (please specify what this means - I assume it is the mean PSD95 expression across tangential axis divided by maximal PSD95 expression encountered) and that the recovery from depression is determined by the radial PSD95 size factor (please specify what this means too). Similarly, the authors assume that the amplitude of facilitation is determined by the tangential SAP102 size factor and the decay time constant of the facilitation by the radial SAP102 size factor. I would ask the authors to explain where these suggested mechanisms come from. Limitations implied by the underlying assumptions have to be discussed in the discussion. In the newly added figure, the authors adjusted the model so that PSD95 affects both the depression and the facilitation parameters while SAP102 affects the AMPA/NMDA ratio. I find little evidence for altered AMPA/NMDA ratio in Levy et al. 2015 - mostly SAP102 knockdown reduced the amplitude of both AMPAR and NMDAR-mediated currents.

REVIEWERS' COMMENTS

Reviewer #3 (Remarks to the Author):

The authors have addressed some of my concerns by referring to articles where STP was shown to be at least partly mediated by post-synaptic mechanisms. I have additional remarks on the newly added description of the computational model.

Point 1

In the "Depression model" and "Fast facilitation" the authors refer to normalized "tangential" and "radial" PSD95 or SAP102 size factors. I understood from their previous reply letter that in each of the 11x11 squares the expression values of PSD95 and SAP102 were used as such instead of calculating the mean radial/tangential expression values. Does the same apply here? If so, please correct.

These issues are addressed in our response to Point 3 below. "Size factors" are addressed in step 1 and "means" in step 2 of our model.

Point 2

In the reply letter the authors stated that the mechanism of how PSD95 and SAP102 affect STP remains unclear, but it seems to me that the authors here have assumed a very specific mechanism in the computational analysis.

The purpose of the model is to investigate the implications of the spatial variation in synapses containing different amounts of PSD95 and SAP102. The electrophysiological parameters (e.g. amplitude of EPSP, amplitudes and time constants of STP), experimentally shown to be affected by up- or downregulation of these proteins, are in the model varied based on amount of protein. As the reviewer points out, a model must be implemented by a specific mechanism. Whereas the precise molecular mechanism whereby these proteins affect electrophysiological properties remains unknown, the effect of varying electrophysiological parameters produces a general mechanism whereby the spatial summation properties can be investigated. Thus, the predictions we make are that variations in protein expression affect how different patterns are processed, temporally and spatially. We have further followed the reviewer's suggestion and added a note on the limitations of our approach (see Point 4 below).

Point 3

Namely, they seem to assume that the amplitude of depression is determined by the tangential PSD95 size factor (please specify what this means - I assume it is the mean PSD95 expression across tangential axis divided by maximal PSD95 expression

encountered) and that the recovery from depression is determined by the radial PSD95 size factor (please specify what this means too). Similarly, the authors assume that the amplitude of facilitation is determined by the tangential SAP102 size factor and the decay time constant of the facilitation by the radial SAP102 size factor.

We have rephrased model descriptions in the main text (Results section 'Pax6 mutation transiently disrupts brain network structure and function' and Methods section 'Computational modeling of synaptic responses') to clarify these issues raised by the reviewer.

Regarding PSD95/SAP102 size factor along the radial/tangential axes. Synaptic properties are defined in two steps. Step one. Our original model presented in Zhu et al. (2018) represents spatial variations in synaptic properties along the radial and tangential axes of the 3-month-old mouse. In our work on the lifespan (Cizeron et al., 2020) we showed that the expression level attains a maximum at this age. This first step is described in the second half of the Methods section 'Computational modeling of synaptic responses'. Here we use data on the size (area) of the synapse along the radial and tangential axes (PSD95/SAP102 size). Several studies using combinations of electron microscopy and electrophysiology or fluorescence labeling of AMPA receptors have shown a strong correlation between size, volume, number of receptors and EPSP amplitude. Furthermore, the term "factor" used in the section 'Computational modeling of synaptic responses' only specifies, as the reviewer suggests, that we are not using the experimental values in nanometers but a relative size normalized in the interval 0-1. This model thus represents the spatial variation of PSD95 and SAP102 along the radial and tangential axes (11x11 squares, Point 1 above) of the 3-month-old mouse.

Step 2, bringing in data from the present study to investigate the impact when expression levels of PSD95 and SAP102 are affected by genotype and age. In the second step, based on data in the present study, and the fact that expression levels are maximal at the age of 3 months, we downscale the values of younger and older age groups according to the "normalization" described. To allow for comparisons between ages and genotypes, we bring for each protein and for axes the data of all ages and genotypes and identify min, max and range (max-min) and scale values accordingly. The rationale behind scaling data based on means (Point 1 above) along the radial or tangential axes (mean calculated over the 4 radial subdivisions or over the 10 tangential subdivisions) was motivated by the observation that over all genotypes, ages, axes and proteins, individual subdivisions displayed a variation including potential "outliers" and using the mean allowed for a more conservative (smoothed) approach.

Point 4

I would ask the authors to explain where these suggested mechanisms come from. Limitations implied by the underlying assumptions have to be discussed in the discussion.

As requested, we have expressed a caveat in the main text (in the Results rather than Discussion since the Discussion does not bring up the simulations):

“The model, which does not require a complete understanding of the molecular mechanisms by which PSD95 and SAP102 regulate AMPA receptors, can address the question of how different temporal patterns are integrated based on the level of synaptic spatial diversity and, specifically, whether this integration is sensitive to the temporal pattern.”

See also Point 2 above for a motivation of why our approach allows us to study the effects of variations in electrophysiological parameters from a relatively general perspective without the need for a detailed molecular model that explains how MAGUK proteins regulate the properties of AMPA receptors.

Point 5

In the newly added figure, the authors adjusted the model so that PSD95 affects both the depression and the facilitation parameters while SAP102 affects the AMPA/NMDA ratio. I find little evidence for altered AMPA/NMDA ratio in Levy et al. 2015 - mostly SAP102 knockdown reduced the amplitude of both AMPAR and NMDAR-mediated currents.

The charge transfer produced by a synapse is related to the area under the curve and not the amplitude. Based on the reductions in amplitudes observed in Lévy et al. (2015), we can see that AMPA-EPSP area is reduced to 55% of its original value and NMDA-EPSP area to 64%, so NMDA has a 56% larger decrement in area compared with AMPA. Over repetitive pulses this difference will summate and the difference in final sum is even greater, particularly since NMDA-EPSPs summate more effectively temporally compared with AMPA. Thus, even if the difference in reduction of the amplitude observed in Lévy et al. may seem small, it is in fact large enough to produce a differential effect.